# Rational design of hairpin RNA excited states reveals multi-step transitions

Ge Han[1] & Yi Xue [1✉]

RNA excited states represent a class of high-energy-level and thus low-populated conformational states of RNAs that are sequestered within the free energy landscape until being activated by cellular cues. In recent years, there has been growing interest in structural and functional studies of these transient states, but the rational design of excited states remains unexplored. Here we developed a method to design small hairpin RNAs with predefined excited states that exchange with ground states through base pair reshuffling, and verified these transient states by combining NMR relaxation dispersion technique and imino chemical shift prediction. Using van't Hoff analysis and accelerated molecular dynamics simulations, a mechanism of multi-step sequential transition has been revealed. The efforts made in this study will expand the scope of RNA rational design, and also contribute towards improved predictions of RNA secondary structure.

---

[1] School of Life Sciences; Tsinghua-Peking Joint Center for Life Sciences; Beijing Advanced Innovation Center for Structural Biology, Tsinghua University, 100084 Beijing, China. ✉email: yixue@mail.tsinghua.edu.cn

RNAs represent a class of molecules that are engaged in diverse biological functions, including transcription, alternative splicing, translation, and degradation[1,2]. Apart from the dominant interests in understanding their structures and functions, the design of synthetic RNAs has received great attention for applications in regulating biological processes such as gene expression[3–5], as well as in developing novel biological devices[6], such as RNA switches[7], thermometers[8], fluorescent probes[9,10], and RNA-based computers[11].

On the other hand, the ever-growing explorations into the structure and function of RNA have underscored the importance of RNA dynamics[12,13]. Similar to proteins, RNA molecules under physiological conditions exhibit extensive conformational fluctuations over a broad spectrum of time scales, often linking to their versatile roles in vivo. Such dynamics can be well understood from the perspective of free-energy landscape[14]. A large number of conformational states pre-exist within different local minima dispersed on the free-energy surface. The interconverting rate between two states is dictated by the height of the energy barrier between them. The conformational states with low free energy are highly populated, referred to as 'ground states' (GSs), while others with high free energy, termed 'excited states' (ESs), are sparsely populated. These ESs, which are otherwise sequestered in the energy landscape, become functional species in response to cellular cues[15]. Although predicting alternative conformational states and associated energy barriers remains a major challenge for computational biologists, recent studies have demonstrated the initial success in the rational design of interconverting multi-state conformations of proteins[16,17]. However, studies of RNA design have thus far focused on the single state, leaving the rational design of RNA alternative conformations, including ESs, conspicuously unexplored.

Predefined RNA excited states will beneficially add to the arsenal available for the design of synthetic RNA devices by providing an additional layer of manipulation. The reshuffling between GS and ES can be easily converted into RNA switches as long as the ES is designed to be stabilized by environmental factors such as metabolites. In this regard, an RNA motif with predetermined ES serves as a fundamental building block for RNA devices. In addition, an autonomously reshuffling RNA can be expanded to a molecular machine with parallel processing capabilities, with workloads allocated to alternative states in different proportions by fine-tuning the population of ES.

Here we designed an array of hairpin RNAs (Supplementary Fig. 1 and Supplementary Table 1) with predefined ESs that exchange with GSs through base-pair shift by one or two nucleotides in register. Such secondary structure reshufflings are quite common in RNAs, and often occur in and around loop motifs or non-canonical base pairs on the microsecond to millisecond timescale[12,18,19]. These ESs evade detection by most biophysical techniques due to their low population and short lifetime. However, the nuclear magnetic resonance (NMR) relaxation dispersion (RD) technique has proven to be particularly powerful in characterizing such transient states at atomic resolution[20–23]. To pursue our goal, we first developed a design protocol to generate candidate RNA constructs using *MC-Fold*[24], a program that has been successfully employed to predict potential RNA ESs in several prior studies[25–28]. Next, we employed both the $^{15}N$ $R_{1\rho}$ experiment[29,30] and the $^1H^N$ CEST experiment[31,32] to probe ESs of the designed RNAs, and verified their secondary structures with the aid of a newly developed imino chemical shift predictor[33]. Finally, by performing RD measurements at varying temperatures, along with accelerated molecular dynamics simulations, we have revealed the details of secondary structure switching through base-pair shift by one or two nucleotides in register.

## Results

**Design of a small hairpin RNA reshuffling through one-nucleotide register.** The GS of an RNA makes a transition to ES either through base-pair rearrangement involving concerted breakage and reformation of multiple base pairs[25–28,34], or through structural changes limited to an individual nucleotide, such as base flipping[35], tautomerization, and ionization[36,37]. Here we focused on the design of RNA ESs arising from base-pair rearrangement within a stem region. To this end, we chose a previously reported hairpin RNA, P5c, as the template; this RNA autonomously interconverts between two states through a single-nucleotide shift in register[34]. A salient feature of the secondary structure of P5c is that multiple Watson-Crick (WC) base pairs and G·U wobbles can readily form upon structural switching (Fig. 1a, left). Inspired by this observation, we designed a type of short hairpin RNA consisting of an invariable GCAA tetraloop and a variable 5-bp stem capped by a G·A mismatch. The sequence of the stem was carefully tuned so that stable base pairs could form upon sliding by one nucleotide along the specified direction (Fig. 1a, right). A dangling guanine was appended to 5′-end to optimize the in vitro transcription (IVT) reaction. We defined this type of secondary structure switching as 'Type 1', to

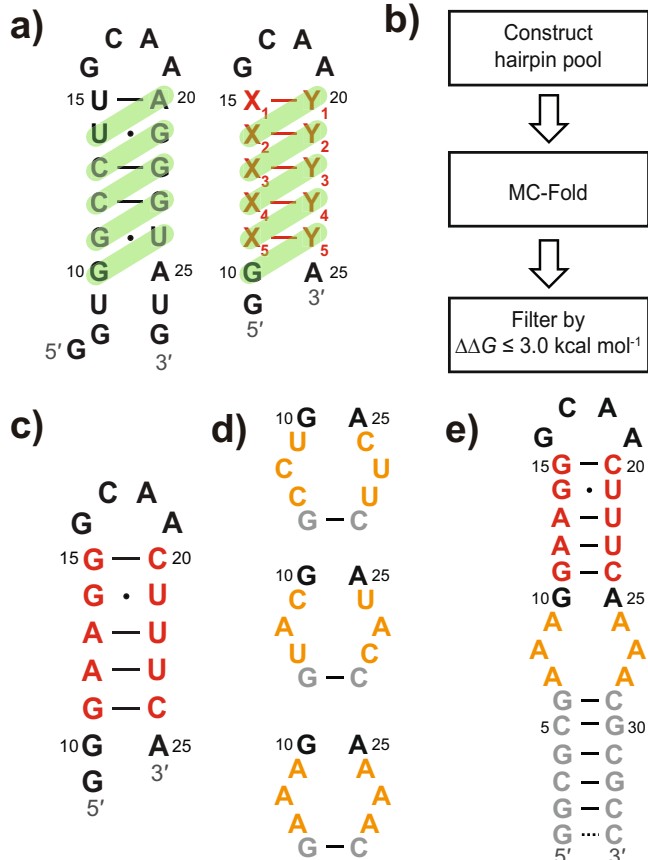

**Fig. 1 Design and verification of T1-short and T1 RNAs. a** Left: The secondary structure of P5c. Base pairs in ES are highlighted by green bars. Right: The design principle of T1-short. Base pairs in GS are colored in red, while those in ES are highlighted by green bars. **b** Schematic design protocol for T1-short. **c** The GS structure of T1-short RNA. The residues in the variable stem are shown in red. **d** Three 3 × 3 internal-loop (orange) candidates. **e** The secondary structure of T1 RNA, which comprises a T1-short hairpin (black and red), a 3 × 3 internal-loop (orange), and a stable lower stem made of alternating G-C and C-G base pairs (gray).

distinguish it from other types of switchings that we designed later.

The design procedure is as follows (Fig. 1b; see Supplementary Methods for details). First, a library of hairpin RNAs was constructed by exhausting all possible combinations of base pairs in the 5-bp stem, provided that the following requirements were met: (1) in the GS, each base pair belongs to 'strong' base pairs (defined as WC base pairs or G·U wobbles); (2) each nucleotide in the stem is centered in a unique base-pair triplet (BP-triplet) to minimize the resonance overlap in the NMR imino 2D spectrum of GS; (3) in the predefined ES formed upon a single-nucleotide register shift (Fig. 1a, shown in green bars), there is up to one 'weak' base pair (defined as G·G, U·U, G·A and A·G non-canonical base pairs), while others are all 'strong' base pairs. Next, *MC-Fold* was used to produce 100 of the most energetically favorable secondary structures for each candidate hairpin. Finally, we used $\Delta\Delta G \leq 3$ kcal mol$^{-1}$ (corresponding to an ES population of ~0.5%, roughly the minimum that is readily detectable by NMR RD experiment) to filter the resulting secondary structure pool associated with each candidate hairpin, where $\Delta\Delta G$ is the free-energy difference between GS and any alternative structure. For a specific candidate hairpin, if the predefined ES falls into the range of $\Delta\Delta G \leq 3$ kcal mol$^{-1}$, and there exists no alternative ES whose $\Delta\Delta G$ is lower than that of the desired ES, then this RNA construct will be accepted as a qualified candidate. The construct with the lowest $\Delta\Delta G$ in the candidate list, termed 'T1-short RNA' (meaning a short version of Type 1 RNA), was chosen for further study.

The NMR imino SOFAST HMQC spectrum of the unlabeled T1-short RNA (Supplementary Fig. 2a) is consistent with the proposed GS structure (Fig. 1c). The secondary structure and the assignment of imino resonances were further confirmed by imino chemical shift prediction and NOESY spectrum of a longer RNA containing T1-short hairpin (Supplementary Fig. 2a, b; see below). To verify the existence of the desired ES, we performed imino $^{15}$N $R_{1\rho}$ experiment on G14 and U21 (Supplementary Table 2) that were expected to undergo G·U → G-C and U·G → U-A transitions, respectively, upon formation of the putative ES. Indeed, we observed $^{15}$N RD signals on these two residues (Supplementary Fig. 2c). The differences in chemical shift between the GS and the ES ($\Delta\omega = \omega_{ES} - \omega_{GS}$) indicate that both G14 and U21 in ES undergo expected downfield shift relative to GS (Supplementary Fig. 2d). Collectively, the data shown above point to the predefined secondary structure reshuffling (Supplementary Fig. 2e).

Despite our success in obtaining an RNA with the desired GS and ES, there remained room for improvement. The yield of IVT for T1-short was poor, likely because of its short length. Given the low concentration of T1-short, we did not proceed to measure RD for other resonances. Instead, we sought to engineer the hairpin to improve transcription efficiency while retaining its ES. Such an optimized construct would allow for additional RD measurements and eventually unambiguous validation of the ES found in T1-short.

It is noteworthy that the modern IVT technique[38] can achieve satisfied yield and purity for short RNAs like T1-short. Considering economy and feasibility, however, we stuck to the conventional IVT, and increased RNA length to improve the efficiency of IVT as well as to reduce other interfering abortive products[39]. In doing so, we appended a stretch of helix to T1-short terminal via a dynamic $3 \times 3$ internal loop. The added lower stem comprises alternating G-C and C-G base pairs, an arrangement that has proven to be thermodynamically stable[40]. Three T1 RNA candidates with different internal loops were selected (Fig. 1d) for the experimental test. As expected, the efficiency of IVT for these samples was significantly boosted. The

imino SOFAST HMQC experiments showed that, among the three RNA constructs, only the one with AAA/AAA internal loop formed the expected secondary structure (Supplementary Fig. 2f), and we termed this construct 'T1 RNA' (Fig. 1e). The spectrum of T1 RNA overlaid well with that of T1-short, except for a few residues near the internal loop (G11, U22, and U23) that showed appreciable offsets (Supplementary Fig. 2a) due to the effect of loop dynamics on chemical shift.

**Verification of the excited state of T1 RNA.** To characterize the ES of T1 with less ambiguity, we carried out a TROSY-based imino $^{1}$H$^{N}$ CEST experiment[32], in addition to a $^{15}$N $R_{1\rho}$ experiment[41]. Consistent with the results from T1-short, we observed RD signals on several residues in the upper stem of T1. Specifically, G11, G14, U21, and U23 showed significant RD signals (Fig. 2a and Supplementary Fig. 3a), G15 and G16 residues that are in or around the apical loop showed moderate RD signals, and U22 showed pronounced $^{1}$H$^{N}$ CEST signal but little to no $^{15}$N $R_{1\rho}$ signal (Supplementary Fig. 3b, c; Supplementary Table 3). The individual two-state fitting gave rise to similar ES population $p_{B}$ (5.7–6.5% ± 0.2–0.5%) and exchange rate $k_{ex}$ (the sum of the forward rate constant $k_1$ and the backward rate constant $k_{-1}$; 420–452 s$^{-1}$ ± 16–44 s$^{-1}$) for residues of G11, G14, U21, and U23, suggesting a concerted exchange process. The global fitting of RD data yielded an ES with a population of $6.2 \pm 0.1\%$ and a lifetime of $2.4 \pm 0.1$ ms (Fig. 2b). No RD signals were detected in residues located in the stable lower stem (Supplementary Fig. 3d).

To verify the designed ES (Fig. 2b), we utilized a newly developed imino chemical shift predictor that can accurately predict $^{15}$N and $^{1}$H$^{N}$ chemical shifts of guanine and uridine according to their BP-triplet context[33]. The predicted values were compared with the experimental chemical shifts of ES that were derived from RD measurements by summing up $\Delta\omega$ and the corresponding chemical shifts of GS (Supplementary Table 3). Four residues (G11, U21, U22, and U23) in T1 are amenable to this approach as they are located in BP-triplets made of WC base pairs or G·U wobbles in ES. Their experimental imino chemical shifts are in good agreement with the predicted values (Fig. 2c). Among these residues, U22 experienced a transition in which the central base pair remained the same, leading to a marginal change in $^{15}$N chemical shift and thereby undetectable $^{15}$N $R_{1\rho}$ signal. Taken together, we successfully obtained a hairpin RNA that reshuffles between two states according to a single-nucleotide register shift. This T1 RNA served as a prototype for the design of other RNAs, as detailed below.

**Stability of the tetraloop modulates reshuffling kinetics of T1 RNA.** Over 50% of all known hairpin loops are tetraloops[42,43], among which GNRA- and UNCG-tetraloops (where N = A, C, G, or U; and R = A or G) account for about 70%[42]. In addition to the GCAA apical loop included in T1 RNA, GAAA and UUCG[44,45] are also among the most frequently occurring tetraloops in RNAs. These tetraloop motifs differ in thermodynamic properties. To investigate the impact of loop stability on the ES, we replaced the GCAA tetraloop of T1 RNA with GAAA and UUCG, respectively, resulting in two constructs: T1-GAAA (Fig. 3a) and T1-UUCG (Fig. 3b), which we analyzed in the same manner as T1.

The imino 2D HMQC spectra of T1-GAAA and T1-UUCG resembled that of T1, except for the characteristic imino peaks arising from their specific tetraloops (Supplementary Fig. 4). The $^{15}$N and $^{1}$H$^{N}$ NMR RD results of these two samples (Supplementary Fig. 5) are consistent with those of T1. Global fitting of RD profiles revealed a population of $6.1 \pm 0.1\%$ and a lifetime of

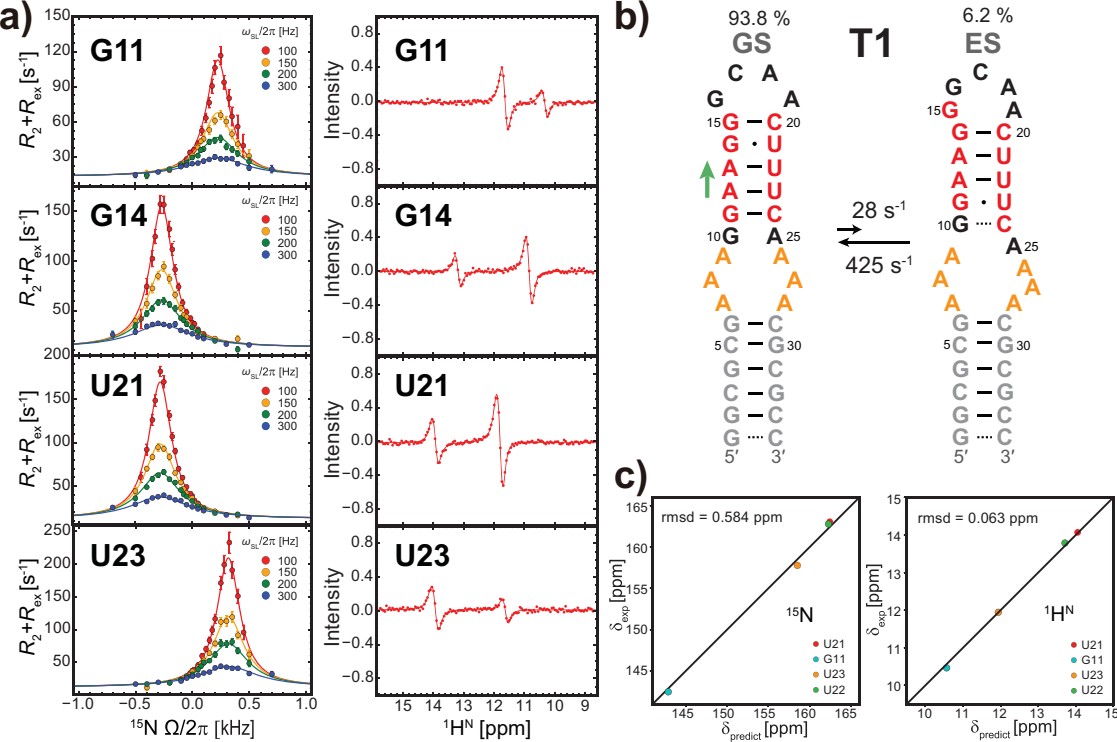

**Fig. 2 Verification of T1 RNA Excited State. a** Representative off-resonance $^{15}$N RD profiles (left panel) and $^{1}$H$^{N}$ CEST profiles (right panel) of residues showing significant RD signals. RD profiles of other residues are available in Supplementary Fig. 3. The error bars in $^{15}$N RD profiles represent standard deviations (SD) estimated using Monte Carlo simulation with 50 iterations. **b** Secondary structures of GS and ES of T1 along with their populations and forward and backward rate constants. Exchange parameters were obtained from the global fitting of RD profiles using a two-state exchange model. The green arrow on the left indicates the direction of sliding. **c** Correlation between predicted and experimental $^{15}$N (left panel) and $^{1}$H$^{N}$ (right panel) chemical shifts of ES for T1.

2.1 ± 0.1 ms for T1-GAAA ES, and a population of 3.3 ± 0.1% and a lifetime of 2.5 ± 0.1 ms for T1-UUCG ES (Supplementary Tables 4–5). The predicted $^{15}$N and $^{1}$H$^{N}$ chemical shifts of ES match very well the experimental values derived from RD results (Supplementary Fig. 5e, j), indicating that both T1-GAAA and T1-UUCG undergo secondary structure reshuffling the same way as T1 (Fig. 3a, b).

Compared to T1 ($k_1 = 28.1 \pm 0.8$ s$^{-1}$), the forward transition of T1-GAAA is slightly accelerated ($k_1 = 30.7 \pm 0.8$ s$^{-1}$), while that of T1-UUCG is slowed down ($k_1 = 13.4 \pm 0.4$ s$^{-1}$). Given that the forward rate constant is dictated by the activation free energy between GS and ES ($\Delta G^{\ddagger}_{GS \to ES}$), which is in turn related to the free energy required to break the tetraloop, the results suggest that the order of stability for these three tetraloops is UUCG > GCAA > GAAA, which is in accordance with prior studies[46,47].

In summary, the data shown above indicate that the stability of tetraloop motifs affects the exchange rate of these autonomously reshuffling RNAs, and the exchange rate between GS and ES can be modulated by changing the tetraloop sequence. In light of these results, we wondered whether the exchange rate could be adjusted by varying the length of the stem where the exchange process occurs.

**Effects of inserting and deleting base pairs on T1 RNA exchange rate.** When sliding of base pairs occurs within a stem region of RNA, multiple base pairs are required to break to allow the new base pairs to form. Straightforward speculation is that elongating or shortening the stem region will alter the activation energy barrier of structural switching, thereby slowing down or speeding up the reshuffling. Indeed, a study on bistable RNAs, which is featured by two comparably populated conformations,

has reported that, for insertion of each additional base pair, the forward rate constant between two states is reduced by 2.5 times[48].

To interrogate the relationship between the exchange rate and the length of the involved stem, we designed T1-delAU and T1-add1bp (Fig. 3c, d; Supplementary Fig. 6) by deleting an A-U base pair or inserting a G·U wobble in the upper stem of T1, respectively. As expected, we detected distinct $^{15}$N and $^{1}$H$^{N}$ RD signals on residues located in the upper stem of T1-delAU (Supplementary Fig. 7a–d) and T1-add1bp (Supplementary Fig. 7f–i). The ESs have been verified by chemical shift prediction (Supplementary Fig. 7e, j and Supplementary Tables 6, 7).

Compared to T1 ($k_{ex} = 453 \pm 13$ s$^{-1}$), T1-add1bp shows a somewhat lower exchange rate ($393 \pm 18$ s$^{-1}$; Fig. 3d), corresponding to an elevation of activation energy by 0.08 kcal mol$^{-1}$. This value is far less than the energy required to break a single hydrogen bond in a WC base pair (~2 kcal mol$^{-1}$)[49], and hence unlikely arises from the inserted base pair. To confirm this, we designed T1-add2bp by further inserting a G·U base pair in the upper stem of T1-add1bp, and the RD measurement informed a slightly decreased exchange rate $k_{ex} = 351 \pm 11$ s$^{-1}$ (Supplementary Figs. 8, 9 and Supplementary Table 8), indicating again an almost unchanged activation energy. Even more unexpectedly, the RD measurement of T1-delAU revealed an exchange process slower than T1 ($347 \pm 10$ s$^{-1}$), despite fewer base pairs required to break upon the transition. Taken together, these results indicate that elongating or shortening the upper stem by one or two base pairs has little effect on the exchange rate.

**Design of RNAs with other reshuffling modes.** All the designed RNA hairpins described thus far undergo GS-to-ES transition

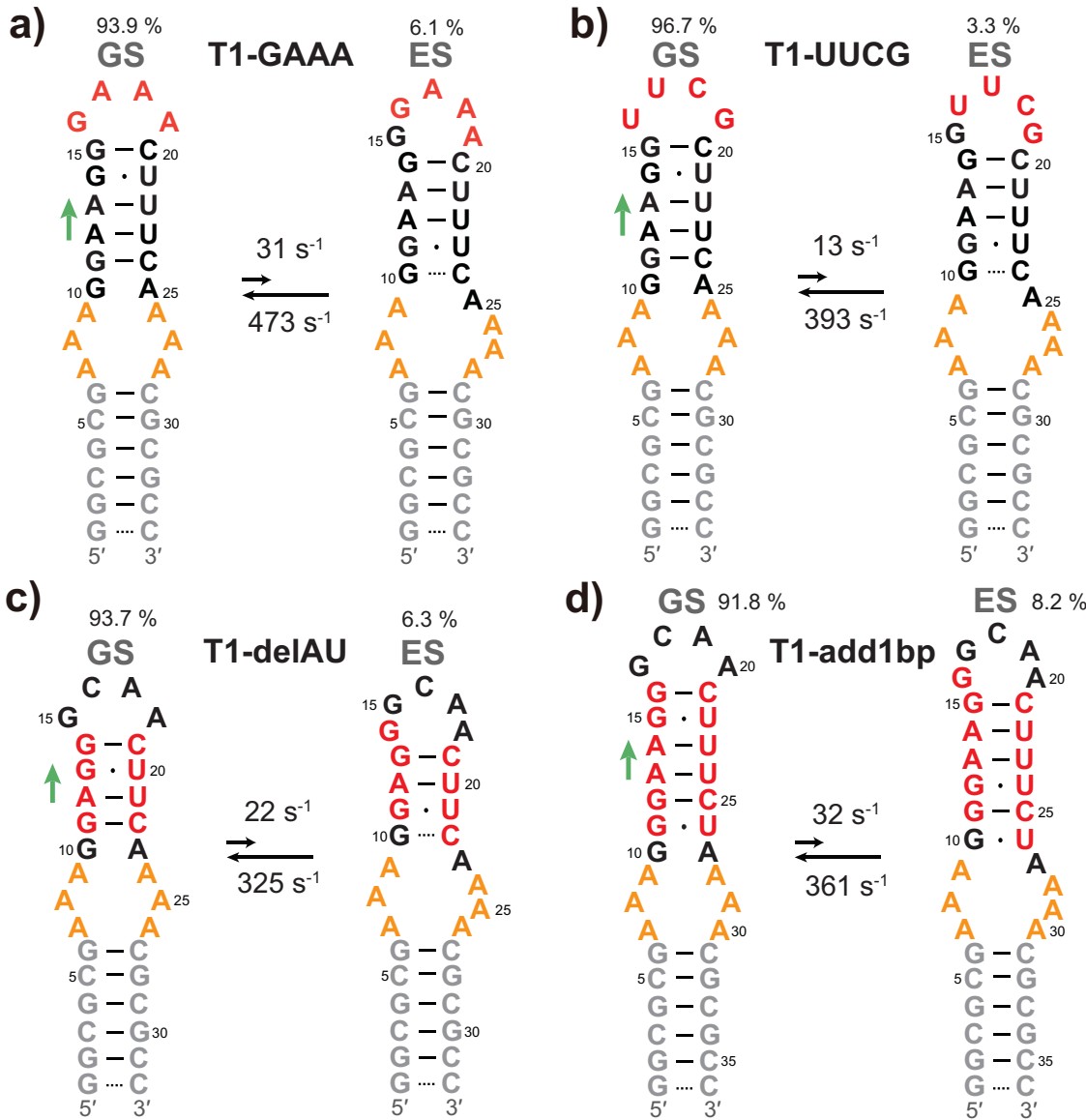

**Fig. 3 T1-GAAA, T1-UUCG, T1-delAU, and T1-add1bp RNAs. a, b** Secondary structures of GS and ES for T1-GAAA (**a**) and T1-UUCG (**b**), along with their populations, and forward and backward rate constants. **c, d** Secondary structures of GS and ES for T1-delAU (**c**) and T1-add1bp (**d**), along with their populations, and forward and backward rate constants. Exchange parameters were obtained from the global fitting of RD profiles using a two-state exchange model. The green arrow on the left of each structure indicates the direction of sliding.

by one-nucleotide sliding from 5′ to 3′. To extend our design to other reshuffling modes, we designed three additional types of RNAs with distinct switching modes: Type 2 RNA, which undergoes a single-nucleotide register shift from 3′ to 5′ (namely, in a direction opposite to that of T1; see Fig. 4a, b); Type 3 RNA, which undergoes a two-nucleotide shift in register from 5′ to 3′ (Fig. 4c, d); and Type 4 RNA, where the structural switching occurs in the lower stem next to a single-bulge element (Fig. 4e, f).

Type 2 RNA was designed using two strategies. The first strategy was to simply swap the positions of the two strands in the upper stem of T1 RNA, leading to T2-mirror RNA (Fig. 4a; Supplementary Fig. 10a, b). From this sample we detected extensive RD signals on nucleotides located in the upper stem (Supplementary Fig. 11a–c). The data show a transition towards an ES with an exchange rate of $580 \pm 39 \text{ s}^{-1}$ and a population of $0.56 \pm 0.03\%$ (Supplementary Table 9). As expected, the resulting $\Delta\omega$ values indicate a secondary structure switching from 3′ to 5′.

The second strategy was to design an entirely new upper stem, resulting in T2 RNA (Fig. 4b, Supplementary Fig. 10c, d and Supplementary Methods). Pronounced RD signals were detected on residues located in the upper hairpin of T2 (Supplementary Fig. 11e–g), revealing an ES with an exchange rate of $426 \pm 15 \text{ s}^{-1}$ and a population of $9.1 \pm 0.2\%$ (Supplementary Table 10). The presumed ESs of these two RNAs were verified by chemical shift prediction (Supplementary Fig. 11d, h). Therefore, we obtained two RNAs that spontaneously reshuffle in the direction opposite to that of T1.

Next, we designed a Type 3 RNA based on a previously identified ES of the transactivation response (TAR) element RNA of HIV-I. In this ES, bulge residues pair up with residues in the upper stem, causing a two-nucleotide register shift that further propagates to the apical loop[26]. The design strategy is illustrated in Fig. 4c, which ensures satisfactory base-pairing upon sliding by two nucleotides (Supplementary Methods). In contrast to the previously discussed RNAs that we designed, where often the first

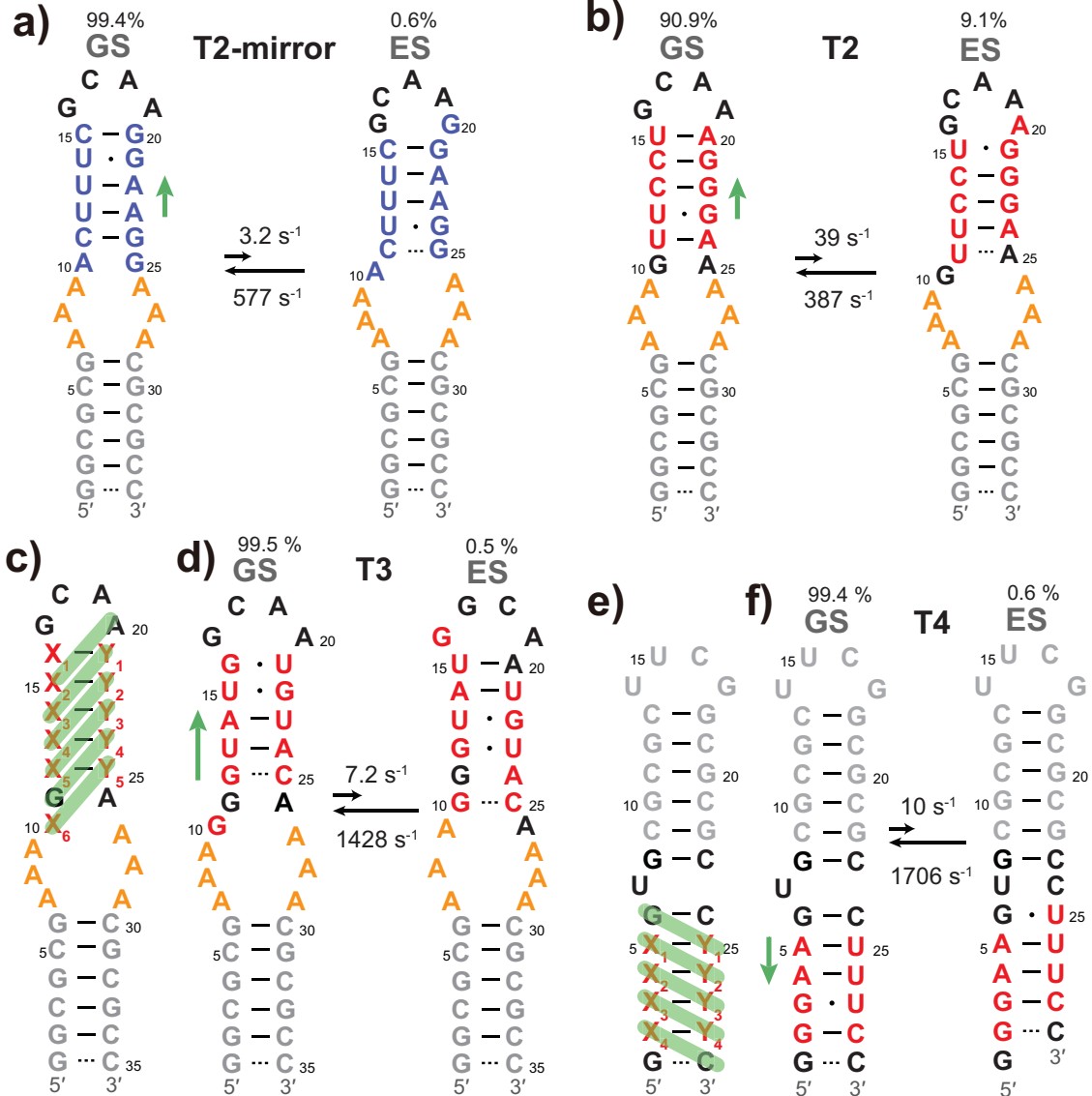

**Fig. 4 T2-mirror, T2, T3, and T4 RNAs. a**, **b** Secondary structures of GS and ES for T2-mirror (**a**) and T2 (**b**) along with their populations, and forward and backward rate constants. T2-mirror and T2 were constructed based on T1 by swapping two strands (blue) or redesigning the sequence (red) of the upper stem, respectively. **c** The design principle of T3 RNA. Base pairs in ES are highlighted by green bars. **d** Secondary structures of GS and ES for T3, along with their populations, and forward and backward rate constants. **e** The design principle of T4 RNA. Base pairs in ES are highlighted by green bars. **f** Secondary structures of GS and ES for T4, along with their populations, and forward and backward rate constants. The green arrow beside each GS structure indicates the direction of sliding.

or second construct that we tested showed the expected GS and ES, more than ten samples were tested (Supplementary Fig. 12a) before we found one that had the correct GS structure (Fig. 4d and Supplementary Fig. 12b, c) and detectable RD signals, namely T3 RNA. We observed the $^{15}$N and $^1$H$^N$ RD signals on several residues localized to the upper stem, including U13, U15, G22, and U23 (Supplementary Fig. 13a–c and Supplementary Table 11). These residues, except G22, experience transitions between WC base pairs and G·U wobbles, thereby serving as sensitive indicators of the reshuffling. The global fitting of RD data revealed a markedly low population of ES with a relatively fast exchange rate ($p_B = 0.50 \pm 0.06\%$, $k_{ex} = 1435 \pm 179\,\text{s}^{-1}$). Of note, owing to the combination of faster exchange and lower population of ES, these RD signals are not as pronounced as those detected in other designed RNAs.

Among residues with readily detectable RD signals, imino chemical shifts of U13$^{ES}$ can be accurately predicted, showing an

excellent agreement with the experimental values (Supplementary Fig. 13d). U15$^{ES}$ and U23$^{ES}$ are not amenable to the imino chemical shift prediction because they are located in the center of a non-canonical BP-triplet. However, their imino chemical shifts (U13$^{ES}$, U15$^{ES}$, and U23$^{ES}$) are in accordance with transitions between U-A and U·G base pairs (Supplementary Fig. 14), despite the large uncertainty in $^1$H$^N$ chemical shift of U15$^{ES}$ due to its weak $^1$H$^N$ CEST signal. Notably, these RD data ruled out the possibility of one-nucleotide sliding in both directions or two-nucleotide sliding in the other direction. For example, U13 forms U·G wobble only in the proposed ES. In all other cases, it forms either U-A canonical base pair or mismatches such as U·U and UC, and is inconsistent with $\Delta\omega$ values derived from RD measurements.

Finally, we tested another type of RNA in which the reshuffling region is placed in the lower stem rather than the upper stem, next to a single bulge separating the two stems (Fig. 4e).

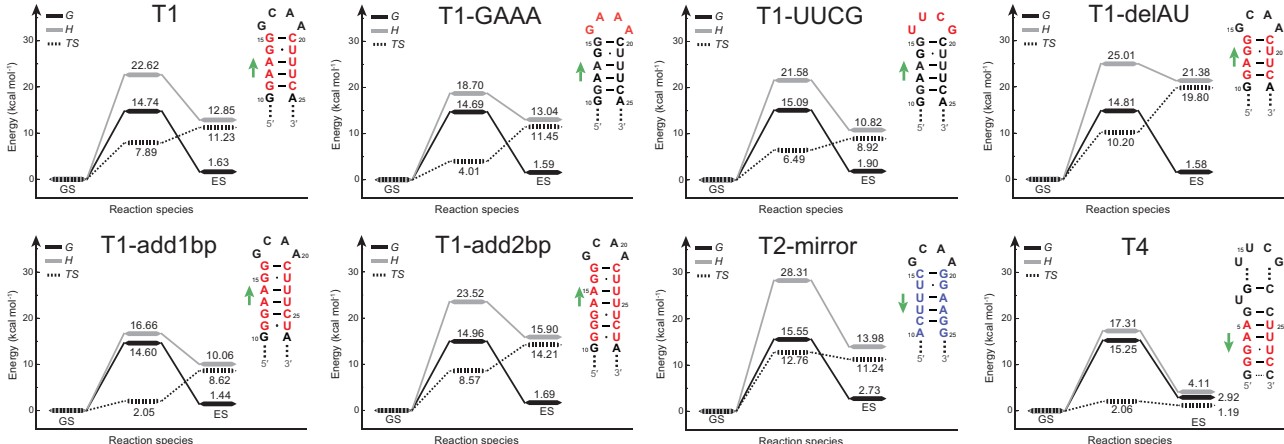

**Fig. 5 Kinetic-thermodynamic profiles of GS-to-ES transitions.** The energy diagrams for the exchange processes between GS and ES via a transition state, in which the activation and net free energy (G), enthalpy (H), and entropy (TS) changes are shown.

Following the similar design pipeline (Supplementary Methods), we obtained a new RNA construct, termed T4 RNA (Fig. 4f Supplementary Fig. 12d, e).

Significant $^{15}$N and $^1$H$^N$ RD signals were observed on residues G3, U25, and U27 within the lower stem of T4 where reshuffling takes place (Supplementary Fig. 13e, f), indicating a sub-millisecond timescale conformational exchange (Supplementary Table 12). In contrast, no RD signals were observed for residues in the upper hairpin region of T4, except for U14 that shows weak RD likely due to local dynamics in the apical loop (Supplementary Fig. 13g). The global fitting gave rise to an exchange process with $p_B = 0.58 \pm 0.02\%$ and $k_{ex} = 1716 \pm 73\,\text{s}^{-1}$, which involves all residues in the lower stem of T4. Again, the secondary structure of ES is supported by the chemical shift prediction (Supplementary Fig. 13h). Thus, we conclude that the lower stem of T4 undergoes a concerted exchange process directed towards the ES as proposed (Fig. 4f).

In conclusion, we successfully designed RNAs with four different reshuffling modes, demonstrating the rationality and generalizability of our design scheme. The design pipeline we used can be easily applied to other reshuffling modes, thus introducing a new dimension into the rational design of RNAs.

**Measurement of the activation energy for secondary structure reshuffling.** We previously concluded that changing the stem length of T1 by one or two base pairs, contrary to our expectations, has little effect on the exchange rates. To understand this unanticipated result, we conducted RD measurements at temperatures ranging from 5 °C to 20 °C on eight RNA constructs, including six T1-derived RNAs, T2-mirror, and T4 (Supplementary Fig. 15; Supplementary Tables 3–9 and 12). T3 was excluded from the measurements due to the weak RD signals. T1 and T1-add1bp RNAs were not measured at temperatures higher than 10 °C, because their imino resonances showing RD signals diminish quickly at higher temperatures. Then we utilized van't Hoff analysis[26,35] to extract a complete set of thermodynamic parameters. Strikingly, the analysis of all these RNAs yielded similar activation free energies (Fig. 5). In addition, similar results were observed in the original design template for T1 RNA, namely P5c ($\Delta G^{\ddagger} \sim 15.1\,\text{kcal mol}^{-1}$; Supplementary Fig. 16).

These uniform activation free energies ($\Delta G^{\ddagger} \sim 14.6$–$15.6$ kcal mol$^{-1}$) and relatively diverse activation enthalpies ($\Delta H^{\ddagger} \sim 16.7$–$28.3$ kcal mol$^{-1}$) match results reported for opening a single WC base pair during the transition to a Hoogsteen base pair in DNA ($\Delta G^{\ddagger} \sim 16$ kcal mol$^{-1}$, $\Delta H^{\ddagger} \sim 12$–$26$ kcal mol$^{-1}$)[35]. Earlier studies have also found that the opening activation energies of

RNA WC base pairs fall in the range of 13–16 kcal mol$^{-1}$ [50]. Therefore, unlike the bistable RNAs where the conformational switch requires disrupting multiple base pairs simultaneously[48], the RNAs designed in this work most likely break only one base pair at each step during the secondary structure transition, which explains why there is no substantial change in $k_{ex}$ upon insertion or deletion of one or two base pairs in the upper stem of T1. It is also noteworthy that the measured activation free energies of T1, T1-GAAA, and T1-UUCG are consistent with their tetraloop stability, as mentioned before.

Although T3 RNA is not suitable for thermodynamic-kinetic analysis due to the low RD signals, a prior study on HIV-1 TAR, where the upper stem also reshuffles through two-nucleotide register shift, showed an activation free energy of 16.8 kcal mol$^{-1}$ [26]. Given the similarity in GS-to-ES switching of the HIV-1 TAR and T3 RNA, it is likely that the transition of T3 experiences a similar activation free energy. Together, our data suggest that the base pair sliding by one or two nucleotides in register is accomplished in multiple steps, with no more than one strong base pair disrupted during each step.

**Visualization of ES-to-GS transitions by accelerated molecular dynamics simulations.** To explore the details of the RNA reshuffling processes, we embarked on accelerated molecular dynamics (aMD) simulations. Compared to conventional molecular dynamics (cMD) simulations whose timescale is often limited to microseconds nowadays, aMD is capable of sampling motions that occur on a timescale of up to milliseconds or slower[51,52]. To increase the occurrence of the structural switch, we started all aMD simulations with ESs rather than GSs (Supplementary Methods).

The first RNA construct for which we successfully observed transitions between ES and GS was T4. For this RNA, we recorded 94 trajectories with lengths ranging from 0.1 µs to 6 µs. In 30 trajectories, we observed 30 ES-to-GS transitions and 2 GS-to-ES transitions. One typical ES-to-GS transition is presented here as the schematic sequential pathway (Fig. 6a and Supplementary Movie 1) and the time course of the base-pairing changes (Supplementary Fig. 17a and c). This transition event was initiated when U25 in the G6·U25 base pair flipped out (A in Fig. 6a). After a duration of 407.01 ns, A5 unpaired with U26 and formed a WC base pair with U25 when the latter flipped back in (B in Fig. 6a). In the following duration of 3.25 ns (B-E), the base pairs located below broke one by one downwards, with the bases on the left (A4, G3, and G2) sequentially pairing with the unpaired bases above (U26, U27, and C28). Afterwards, the two closing base pairs of the lower stem, G6-C24 and G1-C29,

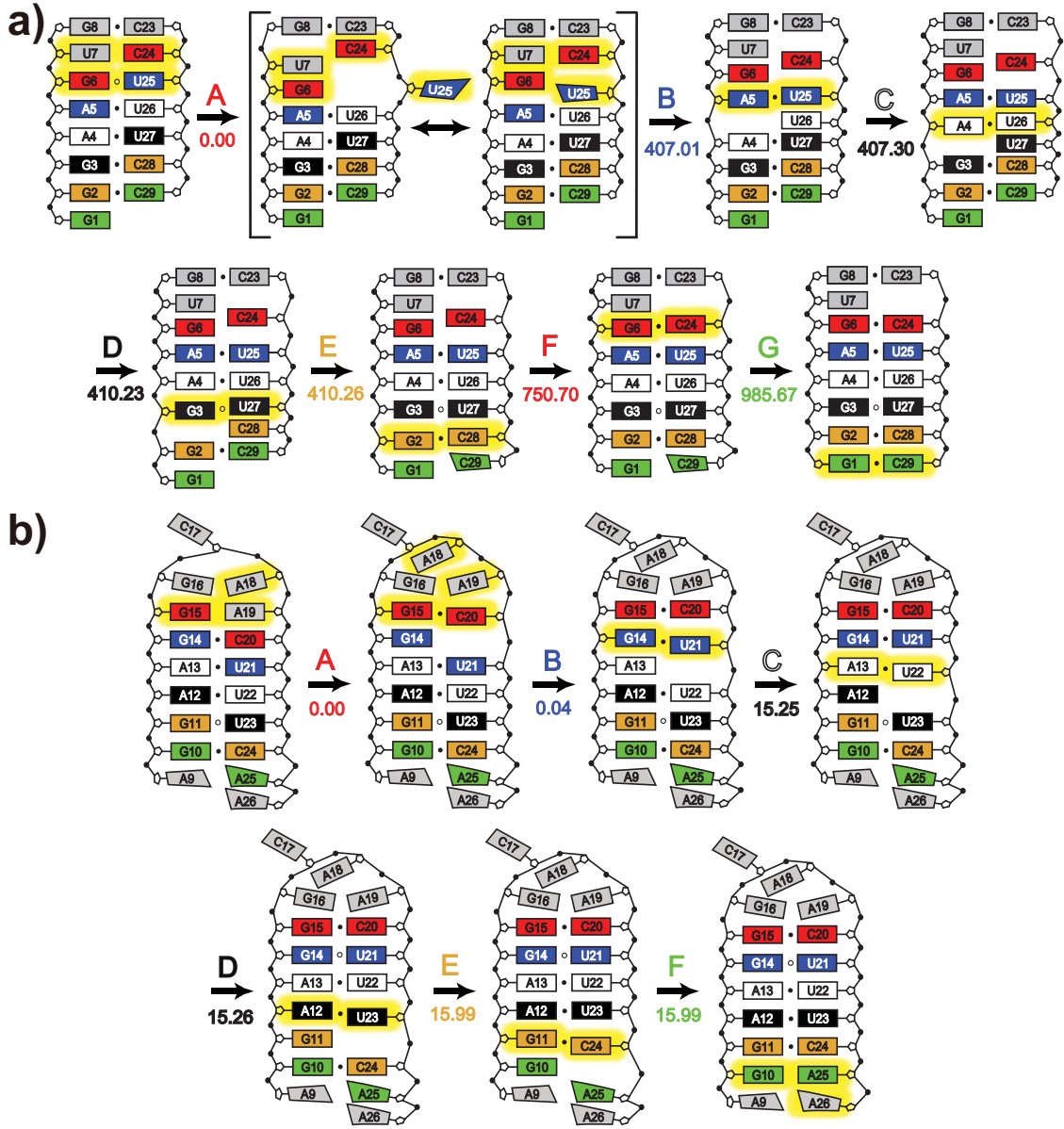

**Fig. 6 The representative ES-to-GS transition pathways. a** A typical downward ES-to-GS transition of T4 RNA. **b** A typical downward ES-to-GS transition of T1 RNA. In each state, residues with status changes relative to the previous state are highlighted in yellow. Each transition point is indicated by a capital letter with a color that matches the color of residues where the major change occurs. The number below each letter represents the time point of state transition in unit of nanoseconds (more precisely, the beginning moment of the state on the right), which can be easily located in the time courses of Supplementary Fig. 17. Please note that the time point of A has been reset to zero.

formed within 575.41 ns (E-G), leading to the GS structure. Such downward transition has been observed 24 times, while the remaining six transitions occurred in an upward manner (Supplementary Movie 2).

We also observed similar transitions in T1 and T2. Among 161 trajectories of T1, three ES-to-GS transitions (Supplementary Movie 3) and one GS-to-ES transition (Supplementary Movie 4) were identified. A typical ES-to-GS transition event started with the opening of G14-C20 located immediately below the pentaloop, followed by a series of one-nucleotide shifts that propagated sequentially downwards (Fig. 6b, Supplementary Fig. 17b, d and Supplementary Movie 3). For T2 RNA, out of 118 trajectories we identified six ES-to-GS transition events, including two downward events (Supplementary Movie 5) and four upward events. Putting together the transition statistics of

trajectories for different RNAs, we found that the ES-to-GS transition occurs much more frequently in T4 than in T1 and T2, which is reasonably consistent with the experimentally measured $k_{-1}$ rates of these three RNAs.

These aMD simulations suggest that the single-nucleotide register shift within a stretch of RNA helix can be divided into multiple steps. Each step involves the disruption of a base pair followed by re-pairing with a nearby unpaired base. This multi-step switching model is consistent with our van't Hoff analysis of NMR RD data. We speculate that a two-nucleotide register shift, such as the reshuffling of HIV-1 TAR or T3 RNA, adopts a similar strategy, given the small activation free energy as revealed by HIV-1 TAR[26]. Indeed, the initial disruption and reformation of base pairs spanning two nucleotides can easily take place in the hexaloop of TAR or the 3 × 3 internal loop of T3.

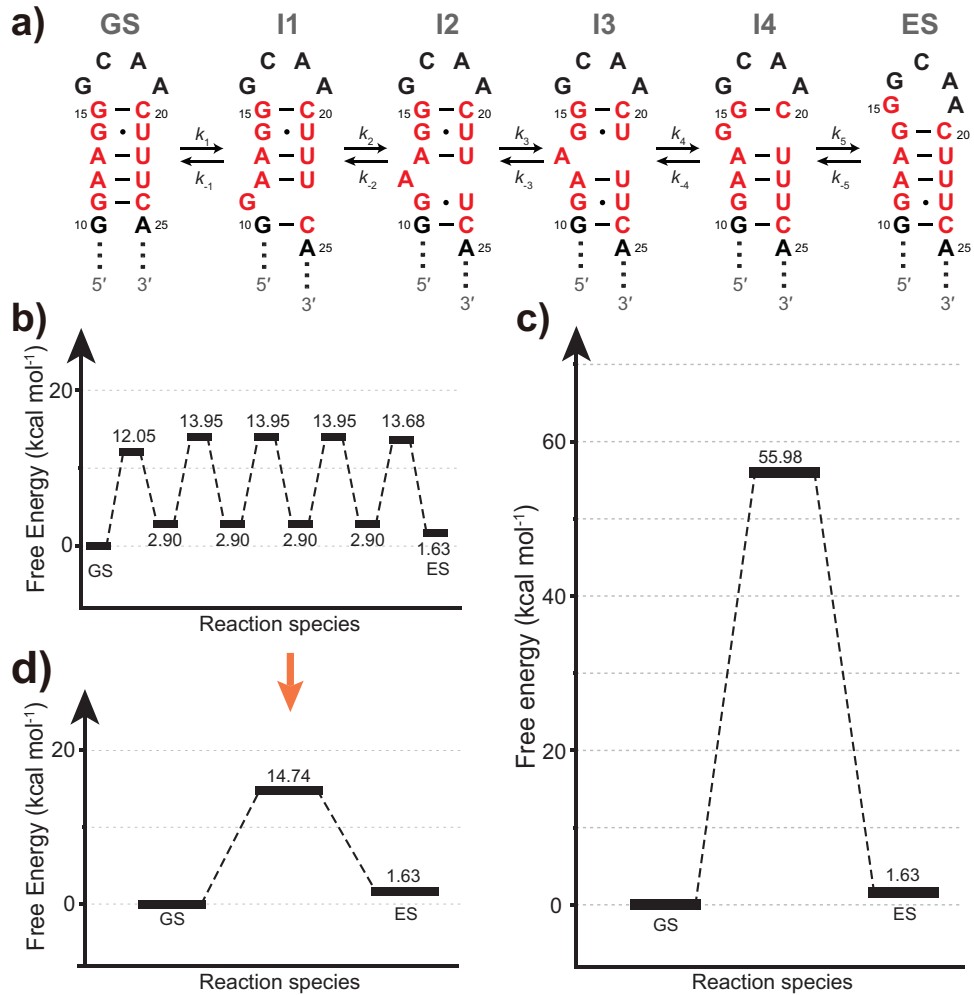

**Fig. 7 Kinetic simulations of T1 RNA. a** The six-state exchange process of an upward GS-to-ES transition of T1 RNA. Shown are the structures of ground state (GS), excited state (ES), and intermediate states (I1–I4). **b** The free-energy diagram of a multi-step transition. The free-energy levels of the four intermediates were elevated above ES by 1.27 kcal mol$^{-1}$, corresponding to a 10-fold lower abundance, to ensure that these intermediates are beyond the detection of NMR RD experiments. **c** The free-energy diagram for the case that all base pairs are broken simultaneously during the transition. **d** The free-energy diagram of the equivalent two-state transition for (**b**).

**Kinetic simulation of T1 RNA reshuffling process**. The multi-step switching mechanism of RNA secondary structure greatly reduces the energy barrier of conformational interconversion, thereby enabling the versatile functions of RNAs. To understand this effect quantitatively, we performed kinetic simulations (Supplementary Methods) by taking T1 RNA as an example, where the transition from GS to ES proceeds according to a six-state upward pathway (Fig. 7a; Supplementary Movie 4). The free-energy difference between GS and ES was fixed to 1.63 kcal mol$^{-1}$, according to the experimental ES population of T1. The heights of five energy barriers were initially set to values comparable to the free energy for disrupting a WC base pair. These barrier heights were then globally fine-tuned until the simulated apparent two-state activation energy exactly matched the experimental value (Fig. 7b, d).

If we assume five base pairs have to be disrupted at the same time, then the energy barrier would be accumulated to a formidable 55.98 kcal mol$^{-1}$ (Fig. 7c), corresponding to an unrealistic forward rate $k_1 = 4.1 \times 10^{-31}$ s$^{-1}$. However, if this high-energy barrier is broken into multiple small barriers (Fig. 7b), the simulated apparent two-state activation free energy becomes 14.74 kcal mol$^{-1}$ (Fig. 7d), only slightly higher than the largest individual barrier (13.95 kcal mol$^{-1}$). This strategy for

reducing the energy barrier may have a broader scope of use for RNAs, rather than being limited to several reshuffling modes discussed here. It is likely through dividing a one-step transition into multiple kinetically favorable substeps, a secondary structure rearrangement is allowed to proceed at a reasonable speed that matches its potential functional role.

## Discussion
The rational design of RNA ground states with diverse functions is thus far a blooming research field[53–55]. In contrast, little attention has been paid to the design of RNA excited states (ESs). Here, with the aid of a secondary structure prediction tool and NMR relaxation dispersion (RD) techniques, we successfully designed and verified a series of artificial RNAs with predefined secondary structure reshuffling directed towards a low-populated ES on the microsecond to millisecond timescale. Through the replacement of apical loop or alteration of stem sequence, we are able to regulate the population of ESs and, to some extent, the transition rates. Our results demonstrate that such design strategy is robust, and is likely applicable to the design of more complex RNA ESs. The ability to engineer an RNA with predefined ESs will expand the range of functionalities that can be fulfilled by synthetic RNA devices, for example, through coupling these

autonomously reshuffling RNAs with existing execution or reporter components of artificial RNA devices[56–58]. In addition to acting as integral components of aptamer, actuator, and transmitter of an RNA switch or other devices[59], a reshuffling RNA with designed switching mode has the potential to perform multiple tasks simultaneously, thereby gaining applications in the context such as alternative splicing[60] or producing miRNA isoforms[61]. Additionally, the impact of environmental factors, such as monovalent or divalent metal ions, on the thermodynamic and kinetic properties of ESs also deserves further investigation.

An interesting question is how different changes made to the RNAs affect the measured exchange parameters (Supplementary Table 1). Comparing T1 with T1-short, we found that an attached $3 \times 3$ internal loop (along with a stable lower stem) significantly influenced both $p_B$ and $k_{ex}$. In contrast, elongating or shortening the stem by up to two base pairs has only a limited effect on these parameters. Next, T1, T1-GAAA, and T1-UUCG differ only in the tetraloop. Such a variation affected the exchange parameters to a small extent, especially for T1 and T1-GAAA whose apical loops differ merely by a single nucleotide. This observation is not surprising, since all the three tetraloops belong to stable GNRA or UNCG motif. Another useful comparison is T2-mirror versus T1, in which swapping two chains of the upper stem led to a significant change in $p_B$. We noticed that T2-mirror[ES] and T1[ES] have different pentaloop sequences, and very likely it is the pentaloop that accounts for the difference in $\Delta\Delta G$. Therefore, tweaking the loop region could be a more effective way to tune exchange parameters than modifying the helical stem.

Van't Hoff analysis of NMR RD data for a number of designed RNAs, including four RNAs with the same type but different stem lengths, indicated that the activation free energy of the secondary structure transition does not depend on the total number of involved base pairs, but is largely equivalent to the energy cost for breaking a single WC base pair. This is in sharp contrast to the case of bistable RNAs where the activation free energy has a linear relationship with the number of base pairs[48]. Based on these results and the insight from aMD simulations, the detailed mechanism of the secondary structure transition involving base pair sliding within a helix has been revealed. Such transitions, involving a shift by one or two nucleotides in register, occur on the microsecond to millisecond time regime, and begin with rare events of a single base pair disruption and reformation in a flexible non-canonical motif located at either end of the helix. The sliding then propagates sequentially until it reaches the other end of the helix. This mechanism is likely applicable to other types of secondary structure transition, as long as the transition pathway can be divided into multiple steps, with each one involving the breakage and reformation of a single base pair. However, for a transition with larger-scale secondary structure rearrangement, multiple base pairs have to break simultaneously to allow the transition to proceed, leading to much slower kinetic rates.

Finally, the growing body of available thermodynamics parameters for RNA ESs, including artificial ESs studied in this work and naturally occurring ones in prior studies, offer an excellent opportunity to improve the current prediction tools of RNA secondary structure. Indeed, these tools, such as *MC-Fold*, failed to predict the free energy of ES to a satisfactory accuracy (Supplementary Fig. 18), calling for more reliable methods. Of note, the mainstream prediction tools make use of thermodynamic parameters derived from optical melting studies, aiming at maximizing the prediction accuracy only for GSs. The NMR-derived thermodynamic properties of ESs will provide these tools with training data in another dimension, thereby further improving the accuracy of predictions.

## Methods

**Sample preparation.** Unlabeled and uniformly $^{13}$C/$^{15}$N-labeled RNA samples were prepared by in vitro transcription using synthetic DNA templates containing the T7 promoter (Genewiz). DNA templates were annealed in 3 mM MgCl$_2$ by heating to 95 °C for 5–10 min and cooling on ice for 30 min. The transcription reaction was carried out at 37 °C for 24 h with in-house purified T7 RNA polymerase using unlabeled (Aladdin) or $^{13}$C/$^{15}$N-labeled nucleotide triphosphates (Cambridge Isotope Laboratories). The samples were purified using 15% (w/v) denaturing polyacrylamide gel electrophoresis (PAGE) in 8 M urea and 1 × TBE buffer, and then eluted by a 'crush and soak' procedure in the corresponding buffer (20 mM Tris-HCl, 0.3 M sodium acetate, 1 mM EDTA, pH 7.4). RNAs were subsequently exchanged into phosphate NMR buffer (10 mM sodium phosphate, 0.01 mM EDTA, pH 6.4) and concentrated to a final volume of about 250 μL using the centrifugal concentrator (3 KDa cutoff). These samples were then refolded by heating at 95 °C for 5–10 min and rapidly cooled down on ice. Prior to NMR experiments, 0.5 μL of 20 mM 4,4-dimethyl-4-silapentane-1-sulfonic acid (DSS) was added to each sample as a chemical shift reference compound, and D$_2$O was added as well to a final concentration of 8% for the purpose of signal locking.

**NMR Spectroscopy and data analysis.** All NMR experiments were performed on Bruker Avance 600 MHz (*Bruker TopSpin* 3.2) or 800 MHz (*Bruker TopSpin* 3.5) spectrometers equipped with a 5 mm triple-resonance TCI cryogenic probe.

*Resonance assignments.* Unless otherwise stated, assignment of imino resonances for all RNA constructs was achieved using a standard set of 2D $^1$H–$^{15}$N SOFAST HMQC and 2D $^1$H–$^1$H nuclear Overhauser effect spectroscopy (NOESY) NMR experiments (acquired using a mixing time of 180 ms at 10 °C). All data were processed and analyzed using software *NMRPipe*[62] and *Sparky* (Goddard, T. D. and Kneller, D. G. SPARKY 3, University of California, San Francisco).

*$^{15}$N $R_{1\rho}$ relaxation dispersion (RD).* Spin-lock powers were calibrated using a modified version of $R_{1\rho}$ pulse sequence[23]. Raw data were processed using *NMRPipe* and *autofit* script to generate a series of peak intensities. On- and off-resonance $R_{1\rho}$ RD profiles with different offset frequencies were recorded under spin-lock powers ($\omega_{SL}/2\pi$) ranging from 100 to 500 Hz (Supplementary Tables 3–12). Magnetization of the spins of interest was allowed to relax under an applied spin-lock field for the following durations: 0–120 ms for N1/N3 in T1-short and T2-mirror, 0–70 ms for N1/N3 in T1 and T3, 0–45 ms for N1/N3 in T1-GAAA, 0–80 ms for N1/N3 in T1-UUCG, 0–35 ms for N1/N3 in T1-add1bp and T1-delAU, 0–50 ms for N1/N3 in T1-add2bp, 0–24 ms for N1/N3 in T2, and 0–100 ms for N1/N3 in T4.

*Analysis of $R_{1\rho}$ data.* $R_{1\rho}$ values were obtained by fitting the decay of peak intensity versus relaxation delay to a mono-exponential curve. Errors in $R_{1\rho}$ were estimated using Monte Carlo simulation with 50 iterations. Measured on- and off-resonance $R_{1\rho}$ data were globally fitted to the Laguerre equation of two-site chemical exchange[63] (Eq. 1) weighted to the experimental error in the $R_{1\rho}$ data.

$$R_{1\rho} = R_1\cos^2\theta + R_2\sin^2\theta + \frac{\sin^2\theta p_{GS}p_{ES}\Delta\omega_{ES}^2 k_{ex}}{\left\{\frac{\omega_{GS}^2\omega_{ES}^2}{\omega_{eff}^2} + k_{ex}^2 - \sin^2\theta p_{GS}p_{ES}\Delta\omega_{ES}^2\left(1 + \frac{2k_{ex}^2(p_{GS}\Delta\omega_{GS}+p_{ES}\omega_{ES}^2)}{\omega_{GS}^2\omega_{ES}^2+\omega_{eff}^2 k_{ex}^2}\right)\right\}}$$
(1)

where, $\omega_{eff}^2 = \Delta\Omega^2 + \omega_{SL}^2$, $\omega_{GS}^2 = (\Omega_{GS} - \omega_{rf})^2 + \omega_{SL}^2$, $\omega_{ES}^2 = (\Omega_{ES} - \omega_{rf})^2 + \omega_{SL}^2$, $\Delta\omega_{ES} = \Omega_{ES} - \Omega_{GS}$, $\theta = \tan^{-1}(\omega_{SL}/\Delta\Omega)$, $\Delta\Omega = \bar{\Omega} - \omega_{rf}$; $R_1$ and $R_2$ are, respectively, the intrinsic longitudinal and transverse relaxation rates, $\Omega_{GS}$ and $\Omega_{ES}$ are the resonance offsets from the spin-lock carrier for the respective states, $\omega_{rf}$ is the reference frequency, $\omega_{SL}$ is the strength of the spin-lock carrier, $\bar{\Omega} = p_{GS}\Omega_{GS} + p_{ES}\Omega_{ES}$, and $k_{ex} = k_1 + k_{-1} = p_{ES}k_{ex} + p_{GS}k_{ex}$ is the exchange rate constant. $^{15}$N RD data were fitted globally by sharing a common set of $k_{ex}$ and $p_B$. The errors in best-fit parameters were estimated using Monte Carlo simulation with 50 iterations.

*$^1$H$^N$ CEST experiments.* All TROSY L-optimized spin state selective $^1$H$^N$ CEST experiments[32] were performed at 10 °C under weak $B_1$ fields and mixing times as shown in Supplementary Table 13. A series of pseudo-3D spectra were acquired by varying the frequency of the $^1$H CEST field from 8.5 ppm through 15.5 ppm in a step-size of 30 Hz. Each 3D spectrum contains two 2D spectra, corresponding to the $N^\alpha$ and $N^\beta$ components of the magnetization, respectively.

*Analysis of $^1$H$^N$ CEST data.* All NMR spectra were processed and analyzed using *NMRPipe*, with peak intensities extracted with the *autofit* script. Analysis of the CEST profiles was carried out using a software package named *ChemEx* (https://github.com/gbouvignies/chemex). The baseline of each CEST profile from $N^\alpha$ or $N^\beta$ component was normalized to 1.0 using a reference 2D spectrum recorded by placing $B_1$ at far-off resonance frequency (−12 kHz). A difference $^1$H$^N$ CEST profile was then produced and fitted to extract chemical shifts of the excited state. During the fitting, $k_{ex}$ and $p_{GS}$ were fixed to values obtained from analysis of $^{15}$N RD data.

**Thermodynamic analysis.** The free-energy change ($\Delta G_i^T$) and enthalpy change ($\Delta H_i^T$) of the activation during a transition were obtained by fitting forward and backward rate constants measured at varying temperatures to the modified van't Hoff equation[64],

$$\ln\left(\frac{k_i(T)}{T}\right) = \ln\left(\frac{k_B \kappa}{h}\right) - \frac{\Delta G_i^T(T_{hm})}{RT_{hm}} - \frac{\Delta H_i^T}{R}\left(\frac{1}{T} - \frac{1}{T_{hm}}\right) \quad (2)$$

where $k_i$ ($i = 1, -1$) represents the forward and backward rate constants of the two-state reaction. $T$ is the temperature in Kelvin. $R$ is the universal gas constant. $k_B$ and $h$ are Boltzmann's constant and Planck's constant, respectively. $\kappa$ is the transmission coefficient (assumed to be 1) in the pre-exponential factor of Eyring's theory, and $T_{hm}$ is the harmonic mean of the experimental temperatures that is computed as $T_{hm} = n / \sum_{i=1}^{n}(\frac{1}{T_i})$.

**Reporting summary.** Further information on research design is available in the Nature Research Reporting Summary linked to this article.

## Data availability

Data have been provided in the main text or the Supplementary Information, including sequence information of RNA constructs that we measured, the protocol and associated parameters of running MD simulations, NMR spectra with resonance assignments, the acquisition parameters and fitting results of NMR relaxation dispersion experiments, and MD movies. The $^1H^N$–$^{15}N$ resonance assignments of the designed RNAs have been deposited in the BMRB database under accession numbers 51238 and 51241-51249. Other data that support the findings of this study are available from the corresponding author upon reasonable request.

## Code availability

The scripts for excited state design and kinetic simulations are available at https://github.com/snowrecall/RNA-design. Other code used to perform calculations of this study is available upon reasonable request to the corresponding author.

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

## Acknowledgements

The authors thank Dr. Ning Xu at NMR facility of the China National Center for Protein Sciences Beijing, for providing facility assistance. We are grateful to Dr. Nikolai Skrynnikov for his helpful comments on the manuscript. This project was supported by funds from the National Natural Science Foundation of China (31770789), the Tsinghua-Peking Joint Center for Life Sciences, and the Beijing Advanced Innovation Center for Structural Biology.

## Author contributions

G.H. prepared samples, and performed all NMR experiments, data analyses, and simulations; Y.X. conceived and supervised the project. The manuscript was written by G.H. and Y.X.

## Competing interests

The authors declare no competing interests.
