## [Peer Review File · Nature Communications]

Title: Rational Design of Hairpin RNA Excited States Reveals Multi-step TransitionsREVIEWER COMMENTS

Reviewer #1 (Remarks to the Author):

Han and Xue designed a number of RNA secondary structure switches, which allowed them to reveal the underlying intermediate states, a question that has been asked a long time: Do complex sec. structure switches have a single high-energy transition state or several, low energy intermediates. The field knew the answer assumed the answer, but Han and Xue for the first time elegantly showed this point, using RD NMR and MD. Showing that the RD exchange rate does not depend on length of the base pair track as well as showing using Van't Hoffs analysis and MD, showing what energy is required, provides a solid experimental bases for this results. This study is thorough and important for development of RNA structure prediction tools and its rational designing for wider application.

Below find some points that could be improved:

Major:

Important: no assignment of T1 besides comparison to T1-short or imino CS predictions (especially issue for the lower stem). The authors cite there previously published work on the imino CS predictions, which though good, have not been tested and validated by others. Therefore at least proper assignment of T1 construct as well as major stem changes (e.g. of T2 mirror, T2, T3 and T4 would be appropriate, also for validation of the imino CS prediction.

Title and abstract do very little to indicate the major conclusion of this work: the elucidation of the base pair switching mechanism. I recommend clarification of title and this sentence, to strengthen this point: Using van't Hoff analysis and accelerated molecular dynamics simulations, a mechanism of multi-step sequential transition has been revealed.

- filtering and analytical scripts, kinetic simulation scripts are not shared to repeat/check work- deposit complete trajectories, so others can check and analyze the data as well
- How many structures were chosen and the pipeline performed on? What does this tell you about the sequence variability? Please provide an overview over all the sequences studied (can be SI). I suggest to keep the labeling constant (e.g. loop has same numbering independent of construct, based on T1 construct)
- T1-short construct doesn't seem to contribute much to the learning outcome – remove for sake of clarity of focus? Or move to SI

Minor:

- Fig. 1A: add numbers to nucleotides, so differences to sec. structure in b) & d) become apparent
- SI design: Step1: are these sequence variation or selection of different ensembles bp steps – please clarify
- SI design: Step3: more details are needed if anyone wants to repeat this (filtering not clear), how many hairpins were left over?

- Why was step 3.4 done? Consecutive identical base-pairs?
- The RD/CEST profiles in the figures should indicate the atom type/number that was measured
- Table of measured parameters and fitted parameters with error indications for individual fits should also be provided, as this can provide a quality measure (please add error for REX parameters in the text)
- For dynamics in T1-short: please indicate GS CS and ES CS in spectra or some other information source, so that one can get an idea on how clear the shift is. (or add a missing reference to second paragraph from the bottom at p5, in case you have already done so) (data similar to table S3 is missing for T1-short- please add)
- Have you considered using solid-phase synthesis or IVT tandem methods for T1-short type of transcripts?
- Modern IVT might not have the same limitations as 1987 (ref 36 for "It has been reported that increasing RNA length improves the efficiency of in vitro transcription and reduces other interfering impurities.³⁶"), please update.
- No data for T1 with CCU/CUU or UAC/UAC bulges, why keep? Or provide data
- Could it be, that T1 CEST G16 and U22 are accidentally mislabeled/switched? If not, why does G16 not have an ES in CEST but U22 have (while the opposite is the case in R1rho)? Please explain
- Instead or in addition of table of sequence (Table S1), a figure of secondary structures next to each other would be more helpful
- Could you report the negative examples of T3 design (SI figure)
- Fig. 6 & S16 are complicated to understand. Not clear: "the vertical axis represents the residue number difference between each residue shown on the left and their paired residue."
- 1. Page 5 line 2: "was used to produce 100 the most..." to "was used to produce 100 of the most..."
- Can the author show a PAGE gel depicting the improvement of the in vitro transcription efficiency for the longer construct compared to the short construct? to support the sentence on page 6 "As expected, the efficiency of in vitro transcription for these samples was significantly boosted"
- In figure S3/S5a/S7a/S9a/S11a can the author comment on the unassigned resonances. The chemical shift suggest them to be from a G in a GC base pair.
- Can the author provide tables with the chemical shift assignment for different constructs?, ideally also deposit in the BMRB

Reviewer #2 (Remarks to the Author):

The manuscript by Han and Xue presents an excellent study in developing an approach to design small hairpin RNAs with predefined conformational transitions. Conformational dynamics plays a critical role in regulatory RNAs. In recent years, it has become increasingly clear that many functional RNA dynamics involve conformational transitions between states with distinct free energies, where the lowest-energy states and alternative states with higher energies are often referred to as ground and excited states, respectively. Since ground and excited states adopt different conformations that can be associated with different functions, it is of significant interests to be able to rationally design RNA sequences that can encode transitions between these distinct functional states, which will have wide applications such as

synthetic biology. However, such a task remains largely unexplored. In the current study, the authors presented a major step forward in this direction. By using secondary structure prediction program MC-Fold, the authors developed a novel approach that utilizes base pair reshuffling as a mode of conformational transitions and successfully designed small hairpin RNA constructs with predefined ground and excited states. A total of eleven designed constructs featuring four different modes of reshuffling were presented, whose dynamic properties were further validated using NMR relaxation dispersion and imino chemical shift prediction. In addition, the authors have carried out temperature-dependent NMR relaxation dispersion measurements, accelerated molecular dynamics simulations, and kinetic simulations to obtain mechanistic insights into the observed conformational transitions, providing strong support for their designing principle of base pair reshuffling. Overall, the study is very well designed, executed, and presented. The approach for designing predefined excited states in small hairpin RNA is innovative and the results are of significant interests to the readers of Nature Communications. Hence, I highly recommend publication of this excellent work.

In the following, I would like to list a few minor points for the authors to consider in improving their manuscript.

1. As a method development work, it would be beneficial to the readers if the authors can provide more specifics and details of their designing process. For example, for designing T1-short RNA, the authors stated that a total of 788 hairpins were generated in step 1. In step 4, the authors stated that qualified sequences from steps 1-3 were compiled into a list for further experimental verification. However, it was not clear how many qualified sequences are there? What is the distribution of $\Delta\Delta G$ s among these qualified sequences? Do these qualified sequences share some common features? Besides the lowest- $\Delta\Delta G$ construct on the list, could the authors provide other criteria for selecting additional qualified sequences for experimental characterization?
2. To overcome the challenge in expressing the short construct, the authors appended a stretch of helix to T1-short terminal through a dynamic internal loop. However, among three constructs, only one construct with an AAA/AAA internal loop forms the expected secondary structure. Since the proposed design platform highly depends on secondary structure predictions, can the authors elaborate on the discrepancy between predicted and experimentally characterized secondary structures of the other two constructs?
3. One important characteristic of conformational transition is the underlying kinetic properties of the ground and excited states. Interestingly, the excited states in T1-short, T3, and T4 RNA constructs all have substantially shorter lifetimes than those in the T1 and T2 RNA constructs. Could the authors provide some insights into these observations?

Reviewer #3 (Remarks to the Author):

Review: NCOMMS-21-28106

Rational Design of Small RNAs with Predefined Excited States

Ge Han, Yi Xue

Overall Recommendation

I recommend this manuscript for publication after minor modifications. As a researcher in the field of RNA structural biology and conformational space, and having read and reviewed other major works on related topics, this is a highly interesting and well-executed study and a great contribution to a growing and exciting field.

General Comments

In the manuscript, "Rational Design of Small RNAs with Predefined Excited States," authors Han and Xue successfully demonstrate and validate, using well-established and reliable methods, the design and characterization of small hairpin RNAs with predefined low-population secondary structures at higher-energy/excited states that predictably reshuffle base-pairing registers by 1 or 2 nucleotides to achieve the ground-state secondary structure. Using NMR-based methods of relaxation dispersion and CEST, and imino chemical shift validation against predicted values, the authors determined the population distribution and exchange rates for a number of constructs with varying design rationale, which included interrogating the effects of tetraloop stability or stem length (Type 1), the direction of reshuffling (Type 2), changes in register of 1 vs 2 nucleotides (Type 3), and location of the reshuffling region (Type 4). They then went on to perform accelerated molecular dynamics simulations using their designed excited-state RNA secondary structures, and successfully demonstrated the mechanisms of nucleotide reshuffling transitions to the ground states, which occur on the timescale of microseconds to milliseconds.

Two major and important conclusions are drawn from the study. 1) the authors successfully fine-tuned and engineered small RNAs that exhibited varying excited-state populations and exchange rates. 2) The combined data reveal a mechanism of sequential intermediates involving the breaking of a single base-pair per each step, as opposed to overcoming a prohibitive activation barrier in disrupting multiple base-pairs simultaneously. These results are likely to have profound and broad implications for the field of RNA biology. As the authors correctly point out, RNA molecules, even the most structured ones, are intrinsically dynamic molecules with complex structure/function relationships. This property, among others, makes RNA structure determination highly elusive, as the molecules exist in an ensemble of conformational states, whose individual subspecies exist as varying population states dictated by free-energy landscapes. As such, our understanding of RNA structure and how it relates to function is severely limited to single lowest-energy states, which are insufficient for understanding the role of molecular dynamics in their relevant and functional contexts. Subsequently, lack of such information impedes rational design of RNAs for biological and biomedical applications, which represents a fast-

accelerating and far-reaching field of research.

The manuscript is exceptionally well-written, thoughtful, and complete. The amount of work that went into performing the experiments, data analysis, and manuscript preparation is obvious and noteworthy. The authors were very thorough and meticulous in presenting their work, and were careful to corroborate and validate their results and conclusions using comprehensive sets of experimental data. The experiments appear to be well executed and their interpretations of data sound. The sheer enormity of the manuscript, particularly the supplemental data, testifies to the diligence and rigor with which the study was conducted. Although the study was convincing enough with NMR data, the supporting MD simulations and accompanying movies add tremendous weight to experimental validation.

Specific Comments

As previously stated, the manuscript is very well written. There were, however, a few typographical errors and a few minor issues with some figures.

1. Page 10, 3rd line from the bottom: "less base pairs" should be corrected to "fewer base pairs."
2. Page 18, 10th line from the bottom: I'm not sure what the authors meant by "acritical." Was this supposed to read "artificial?"
3. Page 18, 8th line from the bottom: "mental ions" should be corrected to "metal ions."
4. Figure 6. This is a great figure that provides static pictures recapitulating the supplementary movies. However:
 - a. It is very dizzying in trying to decipher what changes are being illustrated with each step. I suggest highlighting each specific change at each iteration.
 - b. The color-coded letters A through G, which I assume represent the stages of transition, should be explained briefly in the figure caption. I did not see them referenced in the text, and it is confusing how these correspond to the transitions denoted in Fig. 7a. If left in, one might use numbers instead of letters. Also, it was not clear why the second step in panel A (flipping back in of U25) was skipped in terms of labelling as a step.
 - c. The primary element missing in this figure with respect to the movies is the time domain. It would be very useful and informative to include the time windows/durations associated with each step.
5. Supplementary Figures S5b, S7b, S9d, S11b,d: NOESY spectra are not properly phased.

There are also a few additions that, I feel, would improve the manuscript and its impact on readership.

1. The authors clearly iterate the lack of information with respect to excited/low-population states of

RNA, and how this work enhances the rational design and study of those RNAs. Aside from the obvious implications in aiding in computational studies of RNA dynamics and RNA structure prediction, the manuscript falls short in adequately communicating WHY such information is important and, more explicitly, what potential the information has to offer. This, I believe, is important to reaching a broader audience who may not see the immediate implications. The authors briefly mention the potential use of RNAs with predefined excited states in synthetic RNA devices. This has broader implications of higher significance. The authors should expand this concept further with additional references, and include it in the introduction section. Adding to the impetus for the study and the weight of the implications might better capture the interest of a broader audience.

2. An additional supplementary table of all RNAs used in the study, summarizing the results (k_f , k_r , k_{ex} , populations, free energies, etc.), would be highly beneficial. From this table, the authors may be able to point out and comment on specific trends observed with respect to how different types of changes to the RNAs affect the measured parameters. Which ones are very similar? Which ones are very different? And are those similarities or differences expected by design? As a simple example, on page 9, the authors comment on order of stability of the three tetraloops, based on their observations of forward rate constants. The fact that the k_1 s are so similar for T1-GCAA and T1-GAAA likely results from the loops differing by only 1 nucleotide. Identifying these types of trends, particularly ones of greater complexity, is critical to the future design of RNAs with desired results or effects.

Response to Reviewers

We thank all the reviewers for their careful reviews and insightful comments/suggestions to our manuscript. The manuscript has been carefully revised. The major changes are summarized as follows.

1. The title has been changed into “Rational Design of Excited States of Hairpin RNAs: a Mechanism of Multi-step Sequential Transition is Revealed”.
2. We re-analyzed NMR NOESY data, and obtained the full assignments of all RNAs we measured. These assignments have been provided as Supplementary Figures, as well as a big summary table (Supplementary Tab. 14).
3. All the scripts related to excited state design and kinetic simulations, along with their outcomes and README files, have been provided as a new SI file “scripts_design_simulation.zip”.
4. The subsection “Design of a short RNA that reshuffles autonomously” and the subsection “T1 RNA serves as the prototype for RNA design” have been combined into one subsection. Figure 1 and Figure 2 have been reorganized, and subfigures showing NMR result of T1-short have been moved to SI.
5. More discussions about the potential applications of our work in the design of synthetic RNA devices have been added to *Introduction* and *Discussion*.
6. Figures 1, 2, 6, S1, S2, S3, S5, S6, S7, S8, S9d, S10, S11, S12, S13, and S18 have been updated.
7. References #7, 8, 11, 38, 58, 59, 60, and 61 have been added.
8. Other changes can be found in our detailed response.

Our point-to-point responses to the reviewers’ comments are shown in bold below.

Reviewer #1:

Han and Xue designed a number of RNA secondary structure switches, which allowed them to reveal the underlying intermediate states, a question that has been asked a long time: Do complex sec. structure switches have a single high-energy transition state or several, low energy intermediates. The field knew the answer assumed the answer, but Han and Xue for the first time elegantly showed this point, using RD NMR and MD. Showing that the RD exchange rate does not depend on length of the base pair track as well as showing using Van’t Hoffs analysis and MD, showing what energy is required, provides a solid experimental bases for this results. This study is thorough and important for development of RNA structure prediction tools and its rational designing for wider application.

Specific comments:

Major:

1. Important: no assignment of T1 besides comparison to T1-short or imino CS predictions (especially issue for the lower stem). The authors cite their previously published work on the imino CS predictions, which though good, have not been tested and validated by others. Therefore at least proper assignment of T1 construct as well as major stem changes (e.g. of T2 mirror, T2, T3 and T4 would be appropriate, also for validation of the imino CS prediction.

We totally agree with the reviewer that the complete assignments (including GC/CG-stems) should be provided. Therefore, we re-analyzed all the relevant spectra and obtained the full assignments for all RNA constructs we measured. The NH 2D spectra and NOE walks are shown in Fig S2, S4, S6, S8, S10, and S12. All the assignments have been validated by imino chemical shift prediction.

2. Title and abstract do very little to indicate the major conclusion of this work: the elucidation of the base pair switching mechanism. I recommend clarification of title and this sentence, to strengthen this point: Using van't Hoff analysis and accelerated molecular dynamics simulations, a mechanism of multi-step sequential transition has been revealed.

We thank the reviewer for this very insightful suggestion. The title has been changed into “Rational Design of Excited States of Hairpin RNAs: a Mechanism of Multi-step Sequential Transition is Revealed”.

- a. *Filtering and analytical scripts, kinetic simulation scripts are not shared to repeat/check work-deposit complete trajectories, so others can check and analyze the data as well*

All the scripts related to excited state design and kinetic simulations, along with their outcomes and README files, have been provided (see “scripts_design_simulation.zip” in SI).

- b. *How many structures were chosen and the pipeline performed on? What does this tell you about the sequence variability? Please provide an overview over all the sequences studied (can be SI). I suggest to keep the labeling constant (e.g. loop has same numbering independent of construct, based on T1 construct).*

For the design pipeline, we have provided in this submission all the scripts, the resulting sequence pools, and the final candidate lists (see “scripts_design_simulation.zip” in SI). We examined the sequence variation in each candidate list, and did not find noticeable features. Additional information regarding the number of sequences is also available in our response to Minor Comments #3.

The information of all RNA constructs characterized by NMR have now been provided in Tab. S1 and Fig. S1. The numbering of residues in T1-short RNA has been kept consistent with T1, T2, T3, and T4 RNAs.

- c. *T1-short construct doesn't seem to contribute much to the learning outcome – remove for sake of clarity of focus? Or move to SI.*

We appreciate the reviewer's comment, and agree that the design of T1-short is better seen as a part of the design of T1 RNA. Therefore, we merged the subsection "*Design of a short RNA that reshuffles autonomously*" and the subsection "*T1 RNA serves as the prototype for RNA design*" into a new one "*Design of a small hairpin RNA reshuffling through one-nucleotide register*". The subfigures in Fig. 1 showing the NMR result of T1-short have been moved to SI (Fig. S2c and S2d).

Minor:

1. Fig. 1A: add numbers to nucleotides, so differences to sec. structure in b) & d) become apparent.

Following the suggestion, the nucleotides in Fig. 1a have been numbered.

2. SI design: Step1: are these sequence variation or selection of different ensembles bp steps – please clarify.

In Step 1, taking T1-short as an example, we used a script to generate all possible sequences, 7776 (= 6^5) in total, which was then reduced to 788 sequences after filtering with the two criteria of Step 1. We have modified the text of the design protocol in SI to clarify it. In addition, a new SI file ("*scripts_design_simulation.zip*") containing all design scripts and outcomes has been provided.

3. SI design: Step3: more details are needed if anyone wants to repeat this (filtering not clear), how many hairpins were left over?

In the new version, we have modified the design procedure in SI to make it more detailed, and have uploaded the associated scripts and results along with README files. Below is a quick summary.

T1-short: 788 hairpins in the sequence pool after step 1, and 33 remained in the final list;

T1 (for 3x3 loop design): 216 hairpins in the sequence pool after step 1, and 99 remained in the final list;

T2: 485 hairpins in the sequence pool after step 1, and 257 remained in the final list;

T3: 411 hairpins in the sequence pool after step 1, and 94 remained in the final list;

T4: 27 hairpins in the sequence pool after step 1, and 27 remained in the final list;

4. Why was step 3.4 done? Consecutive identical base-pairs?

Thanks for pointing it out. In the design protocol of the revised SI, we added that “The last filter was used not only to reduce the overlap of NMR spectral resonances via increasing the variation of a sequence, but also to prevent the design with a relatively long segment containing repeatedly the same base pair that could promote the base pair slipping in either direction”.

5. The RD/CEST profiles in the figures should indicate the atom type/number that was measured.

Thanks for pointing it out. Indeed, in some subplots of several SI figures, we forgot to label the atom type. In the revised manuscript, we fixed this issue (specifically, Fig. S2, S3, S5, S7, S11, and S13 in the revised SI).

6. Table of measured parameters and fitted parameters with error indications for individual fits should also be provided, as this can provide a quality measure (please add error for REX parameters in the text).

The measured parameters for RD experiments and the individually and globally fitted parameters (with errors) have been summarized in Tab. S2-S12. We included the errors of exchange parameters in the main text.

7. For dynamics in T1-short: please indicate GS CS and ES CS in spectra or some other information source, so that one can get an idea on how clear the shift is. (or add a missing reference to second paragraph from the bottom at p5, in case you have already done so) (data similar to table S3 is missing for T1-short- please add).

Following the suggestion, we have provided a figure (Fig. S2d; also shown below). This figure shows the GS imino resonances of T1-short, together with the imino regions for various base pairs. To verify the proposed secondary structure switching, the ¹⁵N chemical shift changes of G14 and U21 upon GS-to-ES transition are indicated. The fitting results and fitting errors have been summarized in Tab. S2 in the revised manuscript.

8. Have you considered using solid-phase synthesis or IVT tandem methods for T1-short type of transcripts?

Solid-phase synthesis requires isotope-labeled phosphoramidites, which are very expensive. Tandem IVT tandem is another good way to obtain the NMR sample of a short RNA with decent transcription yield, but the procedure is more complicated than the conventional one. We chose the conventional IVT due to its simplicity and the fact that extending the short hairpin can overcome the issue of poor yield.

9. Modern IVT might not have the same limitations as 1987 (ref 36 for “It has been reported that increasing RNA length improves the efficiency of in vitro transcription and reduces other interfering impurities.36”), please update.

Thanks a lot for the suggestion. We have updated the relevant sentence (see Paragraph 3 on Page 6), that reads “It is noteworthy that modern IVT technique can achieve satisfied yield and purity for short RNAs like T1-short.” A new reference (Karlsson, Feyrer, Baronti. & Petzold. *Curr Protoc* 2021) has been added accordingly.

10. No data for T1 with CCU/CUU or UAC/UAC bulges, why keep? Or provide data.

The 1D imino spectra have been provided as Fig. S2f in the revised manuscript, which has been cited in the main text (Paragraph 3 on Page 6).

11. Could it be, that T1 CEST G16 and U22 are accidentally mislabeled/switched? If not, why does G16 not have an ES in CEST but U22 have (while the opposite is the case in R1rho)? Please explain.

We carefully checked the NOESY spectrum, both G16 and U22 were correctly assigned (See the full NOE walks in Fig. S2b). The assignment is also in excellent agreement with imino chemical shift prediction (see the figure below). It is worth mentioning that we

performed imino chemical shift predictions to validate the assignments for all the resonance assignments we have done.

U22 showed no detectable ^{15}N $R_{1\rho}$ signal simply because both U22^{GS} and U22^{ES} are located in canonical UA base pair, which did not produce a chemical shift change in ^{15}N large enough for $R_{1\rho}$ detection. With regard to G16, the excited state of this residue happened to produce a ^{15}N chemical shift quite different from that of G16^{GS} while producing a $^1\text{H}^{\text{N}}$ chemical shift very close to that of G16^{GS} . In the main text (Paragraph 2 on Page 7), we added the following sentence: “Among these residues, U22 experienced a transition in which the central base pair remained the same, leading to a marginal change in ^{15}N chemical shift (-0.23 ppm) and thereby undetectable ^{15}N $R_{1\rho}$ signal.”

12. Instead or in addition of table of sequence (Table S1), a figure of secondary structures next to each other would be more helpful.

Following this suggestion, we have provided Fig. S1, in which the secondary structures of all studied RNAs are depicted.

13. Could you report the negative examples of T3 design (SI figure)?

Following this suggestion, we have shown 1D imino spectra of 12 failed designs, as well as a table containing their sequences and the reasons for failures (Fig. S12a). More information about how these construct candidates were chosen is available in the new SI file “*scripts_design_simulation.zip*”.

14. Fig. 6 & S16 are complicated to understand. Not clear: “the vertical axis represents the residue number difference between each residue shown on the left and their paired residue.”

Thanks for raising these questions. For Fig. 6, we have revised the figure to follow the suggestions of Reviewer #3, and now the readability has been improved significantly. For Fig. S16 (which becomes Fig. S17 in the revised manuscript), We modified the figure caption to make it clearer: “the vertical axis represents the residue number difference

between a residue of interest (shown to the left of y-axis) and another residue that pairs with it. When the residue of interest is unpaired, the residue number difference is set to zero."

15. Page 5 line 2: "was used to produce 100 the most..." to "was used to produce 100 of the most..."

Fixed. Thanks for pointing out the grammar issue.

16. Can the author show a PAGE gel depicting the improvement of the in vitro transcription efficiency for the longer construct compared to the short construct? to support the sentence on page 6 "As expected, the efficiency of in vitro transcription for these samples was significantly boosted"

Poor yield is an often encountered phenomenon during in vitro transcription of short RNAs, which, according to our understanding, is closely related to the challenge of initiation/elongation during the transcription of short RNAs. Below are photos of big PAGE gels we took for the large-scale transcription (24 mL of reaction volume) of T1-short and T1, with the desired bands marked by red arrows.

17. In figure S3/S5a/S7a/S9a/S11a can the author comment on the unassigned resonances. The chemical shift suggests them to be from a G in a GC base pair.

We have assigned almost all imino resonances carefully in the revised manuscript (Fig. S4, S6, S8, S10, and S12). Now the number of unassigned resonances is very limited, and they show up largely in the same region of different spectra, typically the two resonances around ($^1\text{H}^{\text{N}}$ 13 ppm, ^{15}N 149 ppm) and ($^1\text{H}^{\text{N}}$ 12 ppm, ^{15}N 147 ppm), respectively. These resonances likely stem from the inhomogeneity of the sample or from the impurities.

18. Can the author provide tables with the chemical shift assignment for different constructs? ideally also deposit in the BMRB.

We provided the imino chemical shift assignments for all the relevant RNA constructs (Tab. S14) in the revised manuscript. Considering that these assignments are of little value for structural studies, we kind of prefer to putting them in the SI rather than in the BMRB.

Reviewer #2:

The manuscript by Han and Xue presents an excellent study in developing an approach to design small hairpin RNAs with predefined conformational transitions. Conformational dynamics plays a critical role in regulatory RNAs. In recent years, it has become increasingly clear that many functional RNA dynamics involve conformational transitions between states with distinct free energies, where the lowest-energy states and alternative states with higher energies are often referred to as ground and excited states, respectively. Since ground and excited states adopt different conformations that can be associated with different functions, it is of significant interests to be able to rationally design RNA sequences that can encode transitions between these distinct functional states, which will have wide applications such as synthetic biology. However, such a task remains largely unexplored. In the current study, the authors presented a major step forward in this direction. By using secondary structure prediction program MC-Fold, the authors developed a novel approach that utilizes base pair reshuffling as a mode of conformational transitions and successfully designed small hairpin RNA constructs with predefined ground and excited states. A total of eleven designed constructs featuring four different modes of reshuffling were presented, whose dynamic properties were further validated using NMR relaxation dispersion and imino chemical shift prediction. In addition, the authors have carried out temperature-dependent NMR relaxation dispersion measurements, accelerated molecular dynamics simulations, and kinetic simulations to obtain mechanistic insights into the observed conformational transitions, providing strong support for their designing principle of base pair reshuffling. Overall, the study is very well designed, executed, and presented. The approach for designing predefined excited states in small hairpin RNA is innovative and the results are of significant interests to the readers of Nature Communications. Hence, I highly recommend publication of this excellent work.

In the following, I would like to list a few minor points for the authors to consider in improving their manuscript.

1. As a method development work, it would be beneficial to the readers if the authors can provide more specifics and details of their designing process. For example, for designing T1-short RNA, the authors stated that a total of 788 hairpins were generated in step 1. In step 4, the authors stated that qualified sequences from steps 1-3 were compiled into a list for further experimental verification. However, it was not clear how many qualified sequences are there? What is the distribution of $\Delta\Delta G$ s among these qualified sequences?

Do these qualified sequences share some common features? Besides the lowest- $\Delta\Delta G$ construct on the list, could the authors provide other criteria for selecting additional qualified sequences for experimental characterization?

For T1-short RNA, there were 33 sequences that remained in Step 4. The similar information for other designed RNAs has been provided in the revised design protocol of SI. A summary is available in our response to Minor Comments #3 of Reviewer 1.

The distribution of $\Delta\Delta G$ s of the sequences in the final candidate list has been provided in the SI (see *README* in “*scripts_design_simulation.zip*”) for each designed RNA. In brief, the distribution seems highly dependent on the specific RNA, as summarized in the table below.

Construct	# of candidates	Distribution of $\Delta\Delta G$
T1-short	33	31 (2.0 – 2.5 kcal/mol), 2 (0.6 – 0.7 kcal/mol)
T2	257	257 (2.01 – 3.00 kcal/mol)
T3	94	24 (0.33 - 0.99 kcal/mol), 23 (1.01 - 2.00 kcal/mol), 47 (2.01 - 2.98 kcal/mol)
T4	27	27 (1.61 - 1.67 kcal/mol)

We also examined the sequence variation in each candidate list, and did not find noticeable features. For example, the sequence list for T1-short is shown below. These data have been provided in the new SI file “*scripts_design_simulation.zip*”. In addition to choosing candidate sequences with the lowest $\Delta\Delta G$ s, we also took the following factors into consideration when deciding which constructs will be sent for NMR measurements: 1) the resonances should be dispersed well on 2D imino spectra predicted by imino chemical shift predictor; 2) the priority will be given to RNA constructs with GS-to-ES transition involving base-pair switching between GC and GU, or between UA and UG, in order to ensure pronounced NMR relaxation dispersion signals. We have added the sentences above to the end of the design protocol in SI.

Sequence	$\Delta\Delta G$ (kcal/mol)	Sequence	$\Delta\Delta G$ (kcal/mol)
GGGAAGGGCAACUUUCA	0.6	GGUCCUUGCAAAGGGAA	2.29
GGAGGAGGCAAUUCUUA	0.7	GGGAUCUGCAAGGGUUA	2.29
GGGCCUUGCAAAGGGCA	2.05	GGGAAGUGCAAUUUCA	2.3
GGGCCUUGCAAAGGGUA	2.05	GGUUCUUGCAAAGGAAA	2.3
GGGUCUUGCAAAGGGCA	2.06	GGAGGGUGCAAUCUUA	2.31
GGGUCUUGCAAAGGGUA	2.06	GGGACUUGCAAAGGUCA	2.31
GGAGCUUGCAAAGGUUA	2.1	GGGUCUUGCAAAGGAUA	2.31
GGUUCUUGCAAAGGAGA	2.2	GGAGUUUGCAAAGAUUA	2.32
GGAGGUUGCAAAGCUUA	2.21	GGACCUUGCAAAGGGUA	2.34
GGGUUCUGCAAAGGAUA	2.21	GGAUUCUUGCAAAGGGUA	2.35
GGGAGGUGCAAACUUUA	2.22	GGAGAUUGCAAAGUUUA	2.37

GGGAGGUGCAAACUUCA	2.25	GGGAAUUGCAAAGUUUA	2.38
GGGAGUUGCAAGAUUUA	2.27	GGGAAUUGCAAAGUUCA	2.4
GGAGGGUGCAAACCUUA	2.28	GGAACUUGCAAAGGUUA	2.4
GGGUCUUGCAAAGGACA	2.28	GGAAUCUGCAAGGGUUA	2.41
GGGACUUGCAAAGGUUA	2.28	GGAAAUUGCAAAGUUUA	2.43
GGAAGGUGCAAACUUA	2.28		

2. To overcome the challenge in expressing the short construct, the authors appended a stretch of helix to T1-short terminal through a dynamic internal loop. However, among three constructs, only one construct with an AAA/AAA internal loop forms the expected secondary structure. Since the proposed design platform highly depends on secondary structure predictions, can the authors elaborate on the discrepancy between predicted and experimentally characterized secondary structures of the other two constructs?

The 1D spectra of the other two sequences with 3x3 internal loop (T1-C1 and T2-C2), which failed to form the desired secondary structures, are shown below as well as in Fig. S2f. The sequence pool for choosing the internal loop, as well as the filtered result, can be found in the new SI file (“scripts_design_simulation.zip”).

These results reflect the limited accuracy of MC-Fold in predicting secondary structures. Of note, the AAA/AAA-containing hairpin that we finally chose (i.e. T1-RNA) is also in our candidate list, but it only ranked at #17. It is very likely that some of the top 16 sequences in the candidate list also form the correct secondary structure. We were simply unlucky when picking up T1-C1 and T1-C2. To evaluate the performance of MC-Fold, we compared the $\Delta\Delta G$ predicted by MC-Fold and the experimental $\Delta\Delta G$ derived from the population of ES for each measured RNA in this work, as well as for two other RNAs whose ESs have been characterized in the prior studies. It turns out the correlation (Fig. S18; also shown below for convenience) is not impressive at all. We have mentioned this in the *Conclusion* section that (Paragraph 3 on Page 15): “Indeed, these tools, such as MC-Fold, failed to predict the free energy of ES to a satisfactory accuracy (Supplementary Fig. 18), calling for more reliable methods.”

3. One important characteristic of conformational transition is the underlying kinetic properties of the ground and excited states. Interestingly, the excited states in T1-short, T3, and T4 RNA constructs all have substantially shorter lifetimes than those in the T1 and T2 RNA constructs. Could the authors provide some insights into these observations?

Thanks for raising this interesting question. The lifetime of ES is determined by k_{-1} , which, in turn, is dictated by the free energy difference between ES and TS (Transition State). Let us take T4 and T1 as examples. The k_{-1} rates of T4 and T1 are 1706 s^{-1} and 425 s^{-1} , respectively. The difference, when translated into ΔG , is about 0.8 kcal/mol . We currently do not have a handy tool to predict this ΔG from two secondary structures in transition. However, by inspecting our accelerated MD (aMD) simulations, we found that aMD results provided a semi-quantitative interpretation for this ΔG . In this work, we performed aMD simulations for T4, T1, and T2, whose initial conformations were set to the excited states. It turns out that there are 30 ES-to-GS transitions out of 94 trajectories for T4, three ES-to-GS transitions out of 161 trajectories for T1, and six ES-to-GS transitions for T2. These results are reasonably consistent with the experimentally measured lifetimes of ESs. We added the following sentences into the main text (see Paragraph 1 on Page 13): “Putting together the transition statistics of trajectories for different RNAs, we found that the ES-to-GS transition occurs much more frequently in T4 than in T1 and T2, which is reasonably consistent with the experimentally measured k_{-1} rates of these three RNAs.”

Reviewer #3:

Overall Recommendation

I recommend this manuscript for publication after minor modifications. As a researcher in the field of RNA structural biology and conformational space, and having read and reviewed other major works on related topics, this is a highly interesting and well-executed study and a great contribution to a growing and exciting field.

General Comments

In the manuscript, “Rational Design of Small RNAs with Predefined Excited States,” authors Han and Xue successfully demonstrate and validate, using well-established and reliable methods, the design and characterization of small hairpin RNAs with predefined low-population secondary structures at higher-energy/excited states that predictably reshuffle base-pairing registers by 1 or 2 nucleotides to achieve the ground-state secondary structure. Using NMR-based methods of relaxation dispersion and CEST, and imino chemical shift validation against predicted values, the authors determined the population distribution and exchange rates for a number of constructs with varying design rationale, which included interrogating the effects of tetraloop stability or stem length (Type 1), the direction of reshuffling (Type 2), changes in register of 1 vs 2 nucleotides (Type 3), and location of the reshuffling region (Type 4). They then went on to perform accelerated molecular dynamics simulations using their designed excited-state RNA secondary structures, and successfully demonstrated the mechanisms of nucleotide reshuffling transitions to the ground states, which occur on the timescale of microseconds to milliseconds.

Two major and important conclusions are drawn from the study. 1) the authors successfully fine-tuned and engineered small RNAs that exhibited varying excited-state populations and exchange rates. 2) The combined data reveal a mechanism of sequential intermediates involving the breaking of a single base-pair per each step, as opposed to overcoming a prohibitive activation barrier in disrupting multiple base-pairs simultaneously. These results are likely to have profound and broad implications for the field of RNA biology. As the authors correctly point out, RNA molecules, even the most structured ones, are intrinsically dynamic molecules with complex structure/function relationships. This property, among others, makes RNA structure determination highly elusive, as the molecules exist in an ensemble of conformational states, whose individual subspecies exist as varying population states dictated by free-energy landscapes. As such, our understanding of RNA structure and how it relates to function is severely limited to single lowest-energy states, which are insufficient for understanding the role of molecular dynamics in their relevant and functional contexts. Subsequently, lack of such information impedes rational design of RNAs for biological and biomedical applications, which represents a fast-accelerating and far-reaching field of research.

The manuscript is exceptionally well-written, thoughtful, and complete. The amount of work that went into performing the experiments, data analysis, and manuscript preparation is obvious and noteworthy. The authors were very thorough and meticulous in presenting their work, and were careful to corroborate and validate their results and conclusions using comprehensive sets of experimental data. The experiments appear to be well executed and their interpretations of data sound. The sheer enormity of the manuscript, particularly the

supplemental data, testifies to the diligence and rigor with which the study was conducted. Although the study was convincing enough with NMR data, the supporting MD simulations and accompanying movies add tremendous weight to experimental validation.

Specific Comments:

As previously stated, the manuscript is very well written. There were, however, a few typographical errors and a few minor issues with some figures.

1. Page 10, 3rd line from the bottom: “less base pairs” should be corrected to “fewer base pairs.”

Fixed. Thanks for pointing out the grammar issue.

2. Page 18, 10th line from the bottom: I’m not sure what the authors meant by “acritical.” Was this supposed to read “artificial?”

Fixed. Thanks for pointing out the typo.

3. Page 18, 8th line from the bottom: “mental ions” should be corrected to “metal ions.”

Fixed. Thanks for pointing out the typo.

4. Figure 6. This is a great figure that provides static pictures recapitulating the supplementary movies. However:

- a. *It is very dizzying in trying to decipher what changes are being illustrated with each step. I suggest highlighting each specific change at each iteration.*

Thanks for the nice idea. We modified this figure to follow this suggestion (see Fig S6, which is also pasted below for convenience).

- b. *The color-coded letters A through G, which I assume represent the stages of transition, should be explained briefly in the figure caption. I did not see them referenced in the text, and it is confusing how these correspond to the transitions denoted in Fig. 7a. If left in, one might use numbers instead of letters. Also, it was not clear why the second step in panel A (flipping back in of U25) was skipped in terms of labelling as a step.*

It is true that the meaning of each color-coded letter needs to be made clear — it represents the transition between two immediately adjacent states. More specifically, each transition point is indicated by a capital letter with a color that matches the color

of residues where the major change occurs. In the revised manuscript, we described their meanings in the caption of Fig. 6 and S17, and they are referenced when describing Supplementary Movie 1.

Fig. 6b and Fig. 7a do not represent the same process. The former depicts a downward ES-to-GS transition of T1, while the latter depicts an upward GS-to-ES transition of T1. We have updated the caption of Fig. 7 to reflect the relevant information. We still keep using letters rather than numbers, as they are equally good in our option.

Indeed, the absence of labeling in the second step of Fig 6a is confusing. Thanks for bringing it to our attention. We intended to tell the readers that the second conformation and the third conformation belong to the same state in which these two conformers interconvert (i.e. U25 keeps flipping out and back in). In the revised figure, we emphasized this by using a pair of brackets and a double-headed arrow.

- c. *The primary element missing in this figure with respect to the movies is the time domain. It would be very useful and informative to include the time windows/durations associated with each step.*

It is indeed useful to do so. We modified the figure to add information of each time point in unit of nanoseconds (more precisely, the beginning moment of the state on the right), and updated the figure caption accordingly. Please note that the time point of A has been reset to zero for convenience.

5. Supplementary Figures S5b, S7b, S9d, S11b,d: NOESY spectra are not properly phased.

Thanks very much for pointing out this issue. We have re-phased these NOESY spectra (Fig. S6b, S8b, S10d, S12c,e in the revised manuscript), as well as two additional NOESY spectra showing slight phasing issues (Fig. S2b and S6d in the revised manuscript).

Additional Comments:

There are also a few additions that, I feel, would improve the manuscript and its impact on readership.

1. The authors clearly iterate the lack of information with respect to excited/low-population states of RNA, and how this work enhances the rational design and study of those RNAs. Aside from the obvious implications in aiding in computational studies of RNA dynamics and RNA structure prediction, the manuscript falls short in adequately communicating WHY such information is important and, more explicitly, what potential the information has to offer. This, I believe, is important to reaching a broader audience who may not see the immediate implications. The authors briefly mention the potential use of RNAs with predefined excited states in synthetic RNA devices. This has broader implications of higher significance. The authors should expand this concept further with additional references, and include it in the introduction section. Adding to the impetus for the study and the weight of the implications might better capture the interest of a broader audience.

We highly appreciate the reviewer's insightful comments. In response to these comments, we added a paragraph to the *Introduction* section (Paragraph 3 on Page 3): *"Predefined RNA excited states will beneficially add to arsenal available for the design of synthetic RNA devices by providing an additional layer of manipulation. The reshuffling between GS and ES can be easily converted into RNA switches as long as the ES is designed to be stabilized by environmental factors such as metabolites. In this regard, an RNA motif with predetermined ES serves as a fundamental building block for RNA devices. In addition, an autonomously reshuffling RNA can be expanded to a molecular machine with parallel processing capabilities, with workload allocated to different states by tuning the population of ES."*, **and added more references in the first paragraph. In addition, we added the**

following sentences to the *Discussion* section (Paragraph 2 on Page 14): “In addition to acting as integral components of aptamer, actuator, and transmitter of an RNA switch or other devices,⁵⁹ a reshuffling RNA with designed switching mode has the potential to perform multiple tasks simultaneously, thereby gaining applications in the context such as alternative splicing⁶⁰ or producing miRNA isoforms⁶¹.”

2. An additional supplementary table of all RNAs used in the study, summarizing the results (k_f , k_r , k_{ex} , populations, free energies, etc.), would be highly beneficial. From this table, the authors may be able to point out and comment on specific trends observed with respect to how different types of changes to the RNAs affect the measured parameters. Which ones are very similar? Which ones are very different? And are those similarities or differences expected by design? As a simple example, on page 9, the authors comment on order of stability of the three tetraloops, based on their observations of forward rate constants. The fact that the k_f s are so similar for T1-GCAA and T1-GAAA likely results from the loops differing by only 1 nucleotide. Identifying these types of trends, particularly ones of greater complexity, is critical to the future design of RNAs with desired results or effects.

We thank the reviewer for these very helpful comments. Following the suggestion, we have provided a summary table in Tab. S1, which is shown below for convenience.

RNA	Sequence	p_B (%)	k_1 (s ⁻¹)	k_{-1} (s ⁻¹)	$\Delta\Delta G$ (kcal mol ⁻¹)
T1-short	GGGAAGGGCAACUUUCA	0.28	5.6	1990	3.31
T1	GGCGCGAAAGGAAGGGCAACUUUCAAACGCGCC	6.19	28	425	1.57
T1-GAAA	GGCGCGAAAGGAAGGGAAACUUUCAAACGCGCC	6.08	31	473	1.58
T1-UUCG	GGCGCGAAAGGAAGGUUCGCUUUCAAACGCGCC	3.27	13	393	1.92
T1-delAU	GGCGCGAAAGGGGCAACUUCAAACGCGCC	6.25	22	325	1.56
T1-add1bp	GGCGCGAAAGGGAAAGGGCAACUUUCUAAAACGCGCC	8.17	32	361	1.41
T1-add2bp	GGCGCGAAAGGGGAAGGGCAACUUUCUAAAACGCGCC	5.78	20	331	1.60
T2-mirror	GGCGCGAAACUUUCGCAAGGAAGGAAACGCGCC	0.56	3.2	577	2.92
T2	GGCGCGAAAGUUCUGCAAAGGAAAAACGCGCC	9.14	39	387	1.35
T3	GGCGCGAAAGGGU AUGGCAAUGUACAAAACGCGCC	0.50	7.2	1428	3.00
T4	GGGAAGUGCGCGCUUCGGCGGCCUUUC	0.58	10	1706	2.90

First, comparing T1 with T1-short, we found that an attached 3x3 internal loop (along with a stable lower stem) significantly influenced both p_B and k_{ex} . In contrast, elongating or shortening the stem in transition by up to two base pairs has only a limited effect on these parameters, as we have discussed in the main text. Next, T1, T1-GAAA, and T1-UUCG differ only in the tetraloop, which affects the exchange parameters to a small extent, especially for T1 and T1-GAAA whose apical loops differ merely by a single nucleotide. This observation is not surprising, since all the three tetraloops belong to stable GNRA or UUCG motif. Another interesting comparison is T2-mirror versus T1, in which swapping two chains of the upper stem leads to a significant change in p_B . We noticed that T2-mirror^{ES} and T1^{ES} have different pentaloop sequences, and it is very likely the pentaloop that accounts for the difference in $\Delta\Delta G$. Although computation tools such as MC-Fold can be employed to predict $\Delta\Delta G$, the prediction accuracy is still quite far from satisfactory (Fig. S18), not to mention the more challenging prediction of

ΔG^\ddagger . Interestingly, our aMD results can semi-quantitatively interpret the measured k_{-1} rates. Specifically, trajectories of T4 RNA ($k_{-1} = 1706 \text{ s}^{-1}$) showed much more ES-to-GS transitions than those of T1 and T2 ($k_{-1} = 425 \text{ s}^{-1}$ and 387 s^{-1} , respectively), making it possible to develop MD-based approaches to pursue this goal. Finally, according to the discussions above, tweaking the loop region could be a more effective way to tune exchange parameters than modifying the helical stem.

In the revised manuscript, we have added the discussion above to the main text (Paragraph 3 on Page 14 and Paragraph 1 on Page 13).

REVIEWERS' COMMENTS

Reviewer #1 (Remarks to the Author):

The authors have addressed most comments sufficiently, besides providing access to data and scripts. I believe that sharing this information (data via the BMRB and scripts via public accessible repositories, such as GitHub) is vital for reproduction and any impact the study could have and is today a minimum requirement in science.

Reviewer #2 (Remarks to the Author):

The revised manuscript by Han and Xue has addressed my concerns of the original manuscript. Hence, I highly recommend publication of this excellent work, which should be of significant interests to the readers of Nature Communications.

Reviewer #3 (Remarks to the Author):

The authors seem to have satisfactorily addressed all Reviewer concerns in this revision. The additional text, figures, and tables per Reviewer recommendations improves the content and clarity of the manuscript as a whole.

I have only 1 minor criticism. The revised title is very long. To Reviewer 1's original comment, I believe the suggested sentence was for the abstract. To illustrate this in the title, a shorter version might be "Rational Design of Hairpin RNA Excited States reveals Multi-step Transitions."

Response to Reviewers

Reviewer #1:

The authors have addressed most comments sufficiently, besides providing access to data and scripts. I believe that sharing this information (data via the BMRB and scripts via public accessible repositories, such as GitHub) is vital for reproduction and any impact the study could have and is today a minimum requirement in science.

We have deposited the resonance assignment data in BMRB database under accession numbers: 51238 and 51241-51249. Besides, the scripts and results for excited state design and kinetic simulations have been made available at GitHub. (<https://github.com/snowrecall/RNA-design>)

Reviewer #2:

The revised manuscript by Han and Xue has addressed my concerns of the original manuscript. Hence, I highly recommend publication of this excellent work, which should be of significant interests to the readers of Nature Communications.

Reviewer #3:

The authors seem to have satisfactorily addressed all Reviewer concerns in this revision. The additional text, figures, and tables per Reviewer recommendations improves the content and clarity of the manuscript as a whole.

I have only 1 minor criticism. The revised title is very long. To Reviewer 1's original comment, I believe the suggested sentence was for the abstract. To illustrate this in the title, a shorter version might be "Rational Design of Hairpin RNA Excited States reveals Multi-step Transitions."

We appreciate the reviewer's valuable suggestion. The title has been changed into "Rational Design of Hairpin RNA Excited States Reveals Multi-step Transitions".